# Neural Wave Equations for Irregularly Sampled Sequence Data

**Arkaprava Majumdar[1], M Anand Krishna[1], and P.K. Srijith[1]**

[1]Indian Institute of Technology, Hyderabad
ai24mtech02002@iith.ac.in    cs22mtech14003@iith.ac.in
srijith@cse.iith.ac.in

## Abstract

Sequence labeling problems arise in several real-world applications such as healthcare and robotics. In many such applications, sequence data are irregularly sampled and are of varying complexities. Recently, efforts have been made to develop neural ODE-based architectures to model the evolution of hidden states continuously in time, to address irregularly sampled sequence data. However, they assume a fixed architectural depth and limit their flexibility to adapt to data sets with varying complexities. We propose the neural wave equation, a novel deep learning method inspired by the wave equation, to address this through continuous modeling of depth. Neural Wave Equation models the evolution of hidden states continuously across time as well as depth by using a non-homogeneous wave equation parameterized by a neural network. Through d'Alembert's analytical solution of the wave equation, we also show that the neural wave equation provides denser connections across the hidden states, allowing for better modeling capability. We conduct experiments on several sequence labeling problems involving irregularly sampled sequence data and demonstrate the superior performance of the proposed neural wave equation model.

## 1   Introduction

Sequence data arise in several real-world applications like health care, robotic systems, and speech recognition. Models such as Recurrent Neural Networks (RNNs)( Rumelhart et al. (1986); Hochreiter & Schmidhuber (1997); Cho et al. (2014); De Brouwer et al. (2019); Schuster & Paliwal (1997))and their variants have proven to be highly effective in processing such sequential data. Traditionally, RNNs are perceived as discrete approximations of underlying dynamical systems, a concept well documented in the literature ichi Funahashi & Nakamura (1993); Bailer-jones et al. (2002). However, RNNs face significant challenges in effectively addressing sequence labeling problems that arise in applications such as healthcare, social media, and business, which involve irregularly sampled or partially observed sequence data Rubanova et al. (2019). There have been efforts in the community to develop deep learning models that allow continuous transformation of the hidden representation. Neural Ordinary Differential Equations(Chen et al. (2018)) implicitly model depth by treating it as a continuous transformation of the input-output map. Neural ODEs combine neural networks with ordinary differential equations to achieve this, resulting in an architecture similar to Resnets( He et al. (2016)). This continuous modeling allows for flexible, adaptive representations that capture hierarchical relationships without relying on fixed depth.

Recognizing the limitations of non-uniform data sampling, there has been a paradigm shift toward developing sequence models inspired by Neural ODEs that emulate the continuous evolution of hidden states over time. ODE-RNN(Rubanova et al. (2019)) modeled hidden state transformations over time using a NODE, where hidden representations are continuously transformed taking into account the time gaps between observations, leading to better hidden state representations. Variants of ODE-RNN like the GRU-ODE(De Brouwer et al. (2019)) and ODE-LSTM(Lechner & Hasani (2020)) were consequently proposed for irregular time series data.

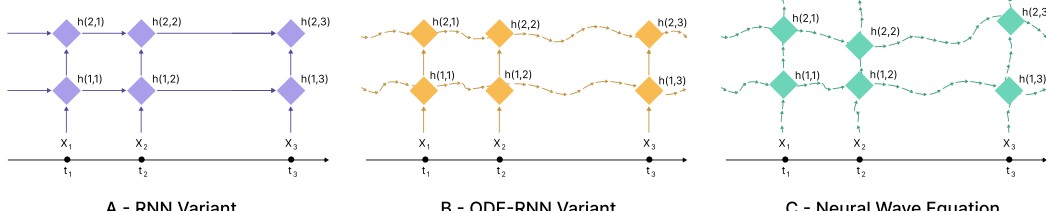

Figure 1: Architectural comparison between discrete depth discrete time, discrete depth continuous time and continuous depth continuous time model

In several real-world problems with irregular observations such as social media post-classification, input data could also exhibit varying complexities, and fixed discrete transformations on the depth dimension would become a limitation. A shallower network may not be able to capture the complexity of the data properly, while a deeper network may overfit the data. ODE- RNN or their variants perform a discrete transformation of the hidden state along depth using neural network transformation, limiting their flexibility to adapt to complex data sets and require exhaustive model selection.

Unifying the principles of Neural ODE and ODE-RNN naturally leads to Partial Differential Equations(Farlow (1993)), which model hidden states continuously over both time and depth. PDEs provide a principled framework for capturing multidimensional dependencies, enabling adaptive representations for hierarchical and temporal complexities.

The continuous depth recurrent neural differential equation Anumasa et al. (2023)(CDR-NDE) proposed the application of a partial differential equation to model the evolution of hidden states continuously over time and depth. The authors of the CDR-NDE paper use the non-homogenous heat equation with the source function being a neural network to model the hidden states continuously over time and architectural depth. Though heat equation-based PDEs are useful for modeling continuous evolution, we find them to have certain limitations that restrict their effectiveness for sequence data. Intuitively, the diffusive nature of the heat equation implies that the initial information is often smoothed out and lost.

We propose the neural wave equation, a wave equation-based neural differential equation, which can provide an effective and natural way to model sequence data. The wave equation can implicitly consider the dependency with neighboring hidden states over a window and addresses the limitations exhibited by the heat equation in sequence modeling. The dependency with some particular hidden states does not arise naturally in the heat equation and hence, needs to be supplemented through the source terms. These dependencies are captured in the wave equation directly. The propagating nature of the wave equation makes it more robust to loss of initial information. Further motivation comes from the existence of an analytical solution for wave equations, and this helps in understanding the effectiveness of wave equations in modeling sequence data. A schematic representation of the hidden state evolution in neural wave equations compared to other popular approaches is provided in Figure 1. Neural wave equations allow for continuous evolution of hidden states across time and depth while capturing more dependencies. We develop neural wave equation models with several parameterizations of the source function through a neural network. Neural wave equations can be solved based on existing solvers for PDE and efficient training techniques based on adjoint methods can be used to learn the parameters. Our experiments, conducted on diverse datasets such as person activity recognition, Walker2d kinematic simulationLechner & Hasani (2020), sepsis (PhysioNet 2019)Reyna et al. (2019) and stance classification Derczynski et al. (2017) demonstrate the superior performance of neural wave equation models over existing baselines for sequence labeling problems. In summary,

1. We propose neural wave equations - a non-homogeneous wave equation with its source function parameterized by a neural network, for sequence labeling problems.

2. We speculate on the potential benefits of using non-homogenous partial differential equations for sequence modeling over continuous time RNN models with discrete depth by looking at their analytical solutions.

3. We empirically demonstrate the effectiveness of the neural wave equations on several sequence labeling tasks with irregularly sampled data.

## 2 RELATED WORK

Several variants of RNNs were introduced in the past to deal with irregularly sampled sequence data or sequence data with missing observations. GRU-D and RNN-D Weinan (2017) models make use of a decay rate for predicting the missing values. CT-LSTMMei & Eisner (2017) combines both LSTM and continuous time neural Hawkes process to model the continuous transformation of hidden states. The latest works in this direction make us of the framework of the neural ordinary differential equation Chen et al. (2018), to model the continuous evolution of hidden states over time.

ODE-RNN(Rubanova et al. (2019)) and their variants such as GRU-ODEDe Brouwer et al. (2019) and ODE-LSTM(Lechner & Hasani (2020)), use a NODE-based formulation to model the hidden state transformation across the irregular observation times, and then an MLP to map it to the corresponding output. Neural Controlled Differential equation (Kidger et al. (2020)) calculates a continuous path over the sequence data using cubic splines and subsequently constructs the evolution of a hidden state in continuous time from it. After a hidden state is calculated, the output from the hidden cell is obtained by passing the hidden state vector through an MLP. Variants of Neural CDE such as attentive Neural CDE(Jhin et al. (2024)) and attentive co-evolving Neural CDE(Jhin et al. (2021)) attempt to combine the attention mechanism with NODE by using two NeuralCDEs. Contiformer (Chen et al. (2023)) introduced a continuous time attention mechanism in transformers (Vaswani et al. (2017)) to model irregularly sampled time-series data. The other promising direction in sequence modeling tasks is the structured state-space models (Gu et al. (2022)) which focuses on discretizing a differential equation with an alternate RNN and CNN view.

Continuous depth recurrent neural differential equations (CDR-NDE) Anumasa et al. (2023) proposed the use of partial differential equations, in particular heat equation, to model the evolution of hidden states over both the temporal and depth dimensions. This overcame the limitations of the discrete depth modeling of the prior approaches for irregularly sampled sequence data. There exists hardly any work on modeling deep learning architectures using PDEs. However, there exists a line of research that aims to use neural networks to solve partial differential equations known as physics-informed neural networks (PINNS) or Neural PDEs Zubov et al. (2021); Brandstetter et al. (2021); Hu et al. (2020); Raissi et al. (2019).

Hughes et al Hughes et al. (2019) draw a similarity between homogeneous wave equation and RNN from a computational physics perspective. In contrast to earlier efforts, the paper focuses on studying the effectiveness of PDEs in developing adaptable deep-learning architectures for modeling, addressing the irregularly sampled sequence data. In particular, we study and propose neural wave equations as an effective solution to solve such sequence labeling problems.

## 3 BACKGROUND

### 3.1 PROBLEM SETTING

We assume a sequence data set consisting of multiple irregularly sampled sequences of length K. Let the elements in the sequence be represented as $\{(t_1, \mathbf{x_1}, y_1), (t_2, \mathbf{x_2}, y_2), ..., (t_K, \mathbf{x_K}, y_K)\}$ where $t_i \in \mathbb{R}+$ is the observation time stamp and $x_i \in \mathbb{R}^D$ is D-dimensional input observed at time $t_i$. The corresponding output $y_i$ can be a class label for a classification task or a real value for a regression task. Our model considers the sequence of observed inputs $[\mathbf{x_1}, \mathbf{x_2}, \ldots, \mathbf{x_K}]$ and their corresponding times $[t_1, t_2, \ldots, t_K]$ as input and aims to predict the output sequence $[y_1, y_2, \ldots, y_K]$ considering the dependencies among the input elements in the sequence and their observation times. Our aim is to learn a function $f(., \theta)$ which can predict the output sequence given the input sequence and observation times so that it exhibits a good generalization performance on the unseen sequences.

### 3.2 IMPLICIT DEPTH IN DEEP LEARNING

The depth of a neural network is an important hyperparameter that determines the model's capacity to learn complex patterns. However, a deeper model may overfit if the input data does not show complex patterns. Implicit layer depth techniques offer a novel approach to structuring deep learning models by adaptively determining the effective depth during training or inference. Models such as Neural ODEs bypass the constraints of predefined layer depth by defining a continuous transformation of

the input with respect to depth via a differential equation. A neural ordinary differential equation (NODE) offers a continuous approach to deep learning by approximating the input-to-output map using a learnable neural network and a differential equation of the following form.

$$\frac{dh(t)}{dt} = f_\theta(h(t), t), \tag{1}$$

where $h(t_0) = h_0$ is the initial condition which is the input to the model or some transformation of the input, and $f_\theta$ is a learnable neural network with parameters $\theta$. The differential equation is then solved with the help of an adaptive step-size solver, which automatically adjusts the step size with varying input complexity. This alleviates the need for tuning the depth of a network manually.

### 3.3 RECURRENT NEURAL ODE

Recurrent Neural ODEs (ODE-RNN Rubanova et al. (2019)) adapt the neural ODE approach to model irregularly sampled sequence data. In the ODE-RNN framework, the dynamics of the system are modeled using a combination of RNN cell and ordinary differential equations, which describe how the state of the system changes continuously over time. Unlike the neural ODE, ODE-RNN models the hidden state evolution continuously in the temporal dimension while following discrete modeling in depth. Assuming, $h_{k-1}^d$ is the hidden state obtained at the previous timestamp $t_{k-1}$ at some discrete depth $d$. $h_{k-1}^d$ is first passed through a neural ODE to obtain the hidden representation $\hat{h}_k^d$ at time $t_k$.

$$\hat{h}_k^d = ODESolve(h_{k-1}^d, (t_{k-1}, t_k)) \tag{2}$$

where ODESolve calls a numerical solver to solve an ODE as referred in Equation 1. The hidden state $h_k^d$ at time $t_k$ is then obtained by passing $\hat{h}_k^d$ and previous layer hidden representation $h_k^{d-1}$ to an RNN-cell, $h_k^d = \sigma(W_{\bar{h}} h_k^{d-1} + W_h \hat{h}_k^d + b_h)$, where $W_{\bar{h}}, W_h, b_h$ are parameters of the RNN-cell. The final output $\hat{y}_k$ at time $t_k$ is obtained by repeating the hidden representation computation over a user-defined discrete depth in RNN-ODE. ODE-RNN provides a better modeling capability, especially for irregularly sampled sequence data. However, their capability is limited by the modeling of depth discretely. In Fig. 1, we see that in ODE-RNN, depth is modeled explicitly by stacking multiple layers of sequential ODE-RNN cells, similar to traditional RNN architectures. In Neural Wave Equation, there is no explicit concept of depth as seen in traditional neural networks with stacked layers. Instead, the transformations of the hidden states are governed by a partial differential equation,

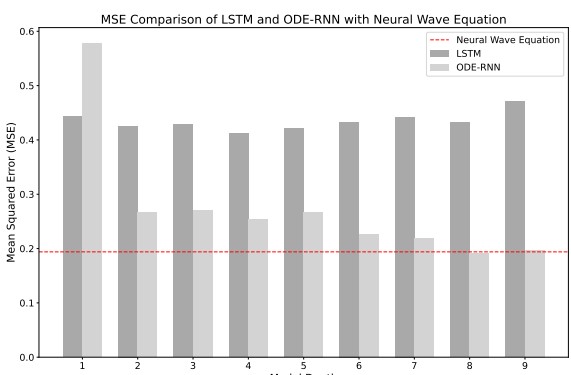

Figure 2: Discrete depth models such as ODE-RNN and LSTM require a model selection over depth to obtain the right model. Neural Wave Equation (red line) achieves this by implicitly and continuously modeling the depth. The dataset used in ETTH1 and the task is to predict 24 time steps in the future while considering the previous 96 timesteps.

and the solver performs a number of small transformations which implicitly define depth (See Section 4). In RNN and ODE-RNN one has to perform exhaustive model selection over depth to achieve a good performance. We demonstrate this in Figure 2 by comparing the performance of ODE-RNN and LSTM for varying depths. We also compare them against the proposed neural wave equation which models depth implicitly (red line).

### 3.4 WAVE EQUATION

In this section, we aim to introduce some ideas and terminologies related to the wave equation which forms the basis of our work. Wave Equation is a partial differential equation of 2nd order in two variables and intuitively, can model more complex dynamics than 1st order PDEs. It describes the propagation of mechanistic waves, such as sound, light, and water waves through a medium. The

propagative nature of the wave equation prevents the loss of initial information as opposed to the heat equation. The homogenous 1D wave equation is formulated as

$$\frac{\partial^2 u(z,t)}{\partial t^2} - c^2 \frac{\partial^2 u(z,t)}{\partial z^2} = 0 \tag{3}$$

with the initial value condition $u(z,0) = f(z), u_t(z,0) = \frac{\partial u}{\partial t}(z,0) = g(z)$ and c is a constant denoting the speed of the wave in the medium. For simplicity, we assume $g(z) = 0$ throughout our work. The solution of the wave equation, $u(z,t)$ gives the displacement of a wave at any given point $z$ over time $t$. It mathematically models how waveforms evolve over time and space in a continuous manner. The analytical solution of wave equations is given by d'Alembert's formula Sobolev as $u(z,t) = \frac{f(z+ct)+f(z-ct)}{2}$. However, we often do not know the explicit form of the initial value function $f(z)$, and instead we know only the values of $f(z)$ at certain points. In such scenarios, we may use a numerical method to solve the wave equation via discretization. One such popular method is the finite difference method(FDM) Abdulkadir et al. (2015). Under the FDM scheme, the wave equation can be discretized as

$$u_{z,t+\Delta_t} = 2u_{z,t} - u_{z,t-\Delta_t} + \frac{\Delta_t^2}{\Delta_t^2} c^2 [u_{z+\Delta_z,t} - 2u_{z,t} + u_{z-\Delta_z,t}] \tag{4}$$

$u_{z,t}$ is the value of the function $u(z,t)$ at position $z$ and time $t$. We can solve the wave equation using the above discretization scheme by using numerical methods for solving ODEs. Higher-order methods like RK-4 are often preferred for higher precision during solving a numerical FDM scheme.

Wave dynamics are better characterized by a non-homogeneous wave equation Farlow (1993) which is written as

$$\frac{\partial^2 u(z,t)}{\partial t^2} - c^2 \frac{\partial^2 u(z,t)}{\partial z^2} = F(z,t) \tag{5}$$

where the function $F(z,t)$ is called a source. It is physically interpreted as an external force that is acting on each point. The source is a function of time and space as well, which means that the external force acting over each data point may vary over time.

## 4 NEURAL WAVE EQUATIONS

Sequence modeling problems require predicting the sequence of outputs while capturing dependencies across the elements in the input sequence. We observe that wave equation offers us a natural way to accomplish this as they are capable of capturing the dependencies and interactions among the system states. We propose to model the evolution of hidden states as a wave equation with the source function parameterized as a neural network. The inputs in the sequence at different observation times are considered as the initial displacement values associated at the various spatial locations in the wave equation. The evolution of wave equation over time implicitly models the depth and number of hidden layer transformations. The proposed neural wave equation captures the dependencies among the hidden states and models their evolution continuously in both the temporal dimension and depth dimension.

Considering the FDM discretization Abdulkadir et al. (2015) for the wave equation in Equation 4. We rewrite it to represent the hidden state evolution with the point $z$ representing the hidden state at some point in time $t$ of the sequence data and time $t$ representing the evolution of the hidden state at some point in depth $d$.

$$h_{t,d+\Delta_d} = 2h_{t,d} - h_{t,d-\Delta_d} + \frac{\Delta_d^2}{\Delta_t^2} c^2 [h_{t+\Delta_t,d} - 2h_{t,d} + h_{t-\Delta_t,d}] \tag{6}$$

Here, $h_{t,d}$ represents the hidden states corresponding to a data point at time $t$ and at depth $d$.

From the FDM expression, we can clearly see the interaction between the hidden states $h_{t,d}, h_{t-\Delta_t,d}, h_{t+\Delta_t,d}$ and $h_{t,d-\Delta_d}$ while calculating $h_{t,d+\Delta_d}$. A drawback with directly adopting a homogenous wave equation in a deep-learning setting is the absence of a learnable function which may be required if the sequence modeling task is complex. To account for it, we also add a learnable neural network to the FDM scheme as

$$h_{t,d+\Delta_d} = 2h_{t,d} - h_{t,d-\Delta_d} + \frac{\Delta_d^2}{\Delta_t^2} c^2 [h_{t+\Delta_t,d} - 2h_{t,d} + h_{t-\Delta_t,d}]$$
$$+ F_{\theta_s}(h_{t-\Delta_t,d}, h_{t,d}, h_{t+\Delta_t,d}, h_{t,d-\Delta_d}) \tag{7}$$

where $F_{\theta_s}(.)$ is a learnable neural network with parameters $\theta_s$. This leads to the proposed neural wave equation model, which is a non-homogeneous wave equation with a source function modeled as the neural network. The neural network $F_{\theta_s}(.)$ helps in capturing the non-linear dependencies with neighboring hidden states. The proposed neural wave equation is given as

$$\frac{\partial^2 h_{t,d}}{\partial d^2} - c^2 \frac{\partial^2 h_{t,d}}{\partial t^2} = F_{\theta_s}(h_{t,d}, h_{t-\Delta_t,d}, h_{t+\Delta_t,d}, h_{t,d-\Delta d}) \tag{8}$$

The initial value condition for the neural wave equation, $f(t_i) = h(t_i, 0)$ for some time $t_i$ is generated from the corresponding input $x_i$ in the input sequence. For simplicity, we assume $\frac{\partial h(t,0)}{\partial d} = 0$. We can get a better understanding of the interaction among the hidden states by studying the analytical solution for the non-homogeneous wave equationSobolev

$$h(t,d) = \frac{f(t+cd) + f(t-cd)}{2} + \frac{1}{2c} \int_0^d \int_{t-c(d-\tau)}^{t+c(d-\tau)} F_{\theta_s}(h_{s,\tau}, h_{s-\Delta_s,\tau}, h_{s+\Delta_s,\tau}, h_{s,\tau-\Delta\tau}) ds d\tau$$
$$\tag{9}$$

where $f(t) = h(t, 0)$ and $F_{\theta_s}(.)$ is the source function. We can see that to compute the hidden state $h(t, d)$ at any time $t$ and depth $d$, it considers the source function values of hidden states in a neighborhood and from the previous depths such as $h(t - \Delta_{cz}, d - \Delta_d), h(t - \Delta_{cz+1}, d - \Delta_d), .., h(t + \Delta_{cz}, d - \Delta_d), h(t - \Delta_{cz}, d - 2\Delta_d), h(t - \Delta_{cz+1}, d - 2\Delta_d), .., h(t + \Delta_{cz}, d - 2\Delta_d)$ and so on where $z$ takes the value of the current depth i.e $d - \Delta_d, d - 2\Delta_d, ...$ A detailed explanation of boundary condition is provided in A.12. We discuss the implications of the analytical solution of the PDEs in general and wave equation in particular in more detail in the following sections.

## 4.1 SOURCE FUNCTIONS

We model the source function terms using a neural network. We use a combination of GRU-Cell and MLPs to model the source terms. The source terms serve two main purposes. It adds non-linearity to the hidden state interactions in the wave equation and provides more control over the flow of information. Secondly, since we are solving a 1-dimensional wave equation with vector-valued hidden states, the source terms enable the mixing of information across the latent dimension of the hidden state. We experiment with several source term formulations of varying numbers of parameters. We experiment with GRU-cells because it provides a gating mechanism to control the flow of information into the concerned hidden state from its neighboring states.

- Single GRU : Model the source term as $F_{\theta_s}(h_{t-\Delta_t,d}, h_{t,d-\Delta_d})$ with only $h_{t-\Delta_t,d}$ and $h_{t,d-\Delta_d}$ passed through a GRU-Cell.

- Single MLP : Models source term as $F_{\theta_s}(h_{t-\Delta_t,d}, h_{t,d}, h_{t+\Delta_t,d}$. All the terms are concatenated together and passed through an MLP layer.

- Double Gating : Models source term as $F_{\theta_s}(h_{t-\Delta_t,d}, h_{t,d}, h_{t+\Delta_t,d}, h_{t,d-\Delta_d})$: The $h_{t,d}$ and $h_{t,d-\Delta_d}$ are concatenated together and passed through a MLP. The output of the MLP is then passed into a GRU-Cell with hidden state $h_{t-\Delta_t,d}$ and into another GRU-Cell with hidden state $h_{t+\Delta_t,d}$ and the results are added.

- MLP+GRU : Models source term as $F_{\theta_s}(h_{t-\Delta_t,d}, h_{t,d}, h_{t+\Delta_t,d}, h_{t,d-\Delta_d})$. The terms $h_{t-\Delta_t,d}, h_{t,d}, h_{t+\Delta_t,d}$ and $h_{t,d-\Delta_d}$ are concatenated and passed into an MLP. The output of the MLP is then passed into a GRU-Cell with hidden state $h_{t,d-\Delta_d}$.

## 4.2 FORWARD PASS

The forward pass in our neural wave equation model consists of 3 stages:

1. An MLP layer that takes the input element in the sequence $x_t$ at time $t$ and projects them into a latent space to obtain the hidden state initial value $h_{t,0}$, $h_{t,0} = MLP_{\theta_{pre}}(x_t)$. Let the initial hidden state values associated with the elements in the sequence be $\mathbf{h}_{:,0}$.

2. The second stage uses an ODESolver to solve the proposed neural wave equation and obtain the final hidden states, $\mathbf{h}_{:,D} = ODESolve(\mathbf{h}_{:,0}, [0, D], ODEFunc)$. ODESolve is a 2nd-order adaptive step size ode solver and ODEFunc is the function that is responsible

for calculating the following discretization term.

$$\frac{\partial^2 h_{t,d}}{\partial d^2} = \frac{1}{\Delta_t^2}[h_{t+\Delta t,d} - 2h_{t,d} + h_{t-\Delta t,d}] + F_{\theta_s}(\cdot) \tag{10}$$

The ODESolve and ODEFunc are used as a black box to solve the neural wave equation using the method of lines. It calculates $h_{t,d}$ for all values of t at once for a particular d and moves forward in depth until it reaches D.

3. The third stage uses an MLP layer that takes the output of the ODESolver and projects it onto the required output space. The output $y_t$ associated with an input $x_t$ and time $t$ is obtained as $y_t = MLP_{\theta_{post}}(h_{t,D})$.

The architecture of the neural wave equation, forward pass and hidden layer interactions can be understood from Figure 3. The FDM method attempts to approximate the analytical solution to a high degree of precision using a particular class of numerical solvers called adaptive step-size solvers Andersson et al. (2015). We used the adaptive step size solver based on Dopri45. It uses the RK-4 and RK-5 as the lower and higher-order solutions respectively. The pseudocode for the algorithm is provided in Appendix A.10. The model parameters, including wave speed $c$, MLP parameters $\theta_{MLP} = (\theta_{pre}, \theta_{post})$ and source function parameters $\theta_s$ are learned using the loss function computed over the output observations in a sequence and over all the sequences. For obtaining the gradients, we use an adjoint sensitivity method developed for PDEs, which works by converting the wave equation to a system of linear 1st-order equations Choon et al. (2019); Lewis et al. (2006).

### 4.3 DISCUSSION

In a normal RNN architecture, the evolution of the hidden state dynamics is as follows: $h_{t,d} = F(W_t h_{t-1,d} + W_{d-1} h_{t,d-1})$. So, the hidden state at point $(t, d)$ depends only on $h_{t-1,d}$ and $h_{t,d-1}$.

In the wave equation, the presence of the integral over the source term from 0 to d ensures that each $h_{t,d}$ is modeled as a function of several hidden states. The trainable parameter c determines the number of the hidden states with depth less than $d$ that contributes to the evolution of $h_{t,d}$.

Most of the other works that combine neural ODE architecture with RNN use the neural ODE to predict the flow of hidden states over a continuous time (Rubanova et al. (2019); Kidger et al. (2020)). However, they are still discrete in the depth direction. CDR-NDE Anumasa et al. (2023) addresses this by using a PDE based on heat equation.

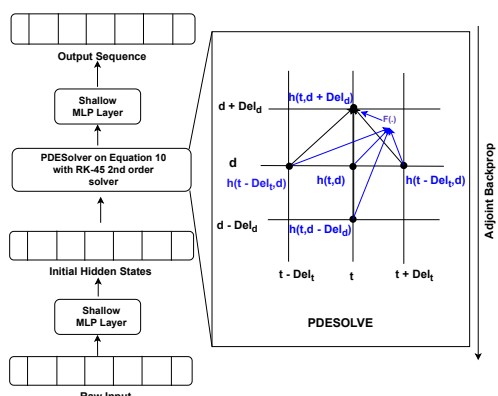

Figure 3: Neural wave equation architecture consists of a shallow MLP over input, PDE solver, and a shallow MLP to produce the output.

During our investigation, we note that the reason PDEs can be used to model sequence data lies in their analytical solution. The analytical solution of $h_{t,d}$ where the evolution is governed by a PDE will often incorporate a term like $\int_0^d \psi(\tau) \int_t F(s, \tau) ds d\tau$. This implies that a particular hidden state at an arbitrary depth is affected directly by all the values of hidden states at a lower depth. Consider the last term in Equation 9, which provides d'Alembert's solution for the wave equation. It considers all the source terms from the previous depths at each time point to compute the hidden state at the current depth. This is also true in the case of the Heat Equation. The analytical solution of the heat equation is given by separation of variable Widder (1976)

$$h(t, d) = \sum_{n=1}^{\infty} (a_n(0) \exp^{(-k\lambda_n d)} + \int_0^d q_n(\tau) \exp^{(-k\lambda_n(t-\tau)} d\tau))\phi_n(t) \tag{11}$$

where $\phi_n(t) = sin\frac{(n\pi t)}{T}$, and $q_n(d) = \int_0^T F(t, d)\phi_n(t)dt$. The presence of the negative exponential term in the solution of the heat equation means that the effect of the hidden states located at lower

depths is diminished while calculating the hidden states located at higher depths. The wave equation does not suffer from this problem as can be observed from its analytical solution.

## 5 EXPERIMENTS

The performance of the proposed Neural wave models was assessed through experiments on datasets containing irregular sequence data such as person activity recognitionMarkelle Kelly (2000), walker2d-v2 kinematic simulationLechner & Hasani (2020), PhysioNet sepsis prediction, and stance classification of social media posts Derczynski et al. (2017). These models are benchmarked against baselines specifically developed for handling irregular sequence data. The experimented configuration includes setting the hidden state dimension to 64 for all source functions, keeping a minibatch size of 256, use of the Adam optimizer, a learning rate of $5 \times 10^{-3}$, and 200 training epochs. These configurations follow the guidelines as established in Lechner & Hasani (2020). The first MLP layer is a single layer with hidden dimension 64. The last MLP layer is also a single layer with hidden dimension equal to the output size. We use the Tsit5 from the package torchdyn Poli et al. as our adaptive solver, which is an efficient reimplementation of the Dopri45 by the Julia Computing group Rackauckas & Nie (2017). The information about the step size and the ODESolvers that have been used for all our models and baselines is mentioned in Table 4 in Appendix A.13. Model training is conducted on an Nvidia Tesla V-100 32GB GPU and an L4 GPU. In our evaluation, we measured the efficacy of our newly developed model against a set of established baselines. These include GRU-ODE, CT-GRU, CT-RNN, GRU-D, Phased-LSTM, ODE-LSTM, bidirectional-RNN, RNN decay, Hawk-LSTM, Augmented LSTM, ODE-RNN, Neural CDE and CDR - NDE models.

### 5.1 RECOGNIZING PERSON ACTIVITY FROM IRREGULARLY SAMPLED TIME-SERIES

The dataset consists of sensor readings from four sensors attached to five individuals (ankle, chest, and belt) performing five activities. The objective is to utilize this sensor data to categorize the performed activities. Initially containing 11 activities, it was refined to 7 classes as recommended by Rubanova et al. (2019). Each recording step includes 7 values, 4 indicating active classes and 3 representing sensor data. Data is segmented into overlapping 32-step intervals with a 16-step overlap, yielding 7,769 training and 1,942 testing sequences. We evaluate our model against established baselines for irregularly sampled activity recognition Markelle Kelly (2000). The neural CDE model achieves $75.16\% \pm 0.71$ accuracy after 40 epochs. While GRU-based models perform best among baselines, they are surpassed by Neural Wave Equation variants. On average, the solver makes 32 function calls in the person dataset and 26 in the walker dataset.

In Table 1, Column 2 presents the test accuracy for all models trained on the person-activity recognition dataset. Notably our **Neural Wave model - Double Gating** variant outperforms all the established baseline models with the highest test accuracy.

### 5.2 WALKER2D-V2 KINEMATIC SIMULATION.

The Walker2D dataset Lechner & Hasani (2020) is derived from simulations in the Walker2d-v2 OpenAI Gym environment, powered by the MuJoCo physics engine. The underlying motion of the walker is governed by continuous-time physical dynamics, simulating kinematic systems evolving smoothly over time. The training set was compiled through rollouts in the Walker2d-v2 environment under a deterministic policy pre-trained via Proximal Policy Optimization, albeit employing a non-recurrent policy framework. To achieve irregular sampling, 10% of the timesteps were omitted. The data is partitioned into 9,684 training sequences, 1,937 for testing, and 1,272 for validation.

We tested the performance of our model on irregularly sampled Column 3 of Table 1 delineates the efficacy of various models on the Walker2d dataset. Our proposed **Neural Wave - Single MLP** model outperforms all the baselines. We were unable to run the Neural CDE modelKidger et al. (2020) due to the long duration required to complete one epoch. We suspect that the construction of the continuous path with cubic splines is a bottleneck in the Neural CDE model, as increasing the sequence length and dimension of input features significantly slows it down. Even in the Person's activity dataset, the neural CDE model took 300 sec compared to 18-30 seconds by that of neural wave equation or 30 - 50 secs of CDR-NDE models. Computational complexity is discussed in detail in A.7.

Table 1: Column 2 outlines the test accuracy (mean ± standard deviation) of each model trained on the dataset titled **Person Activity Recognition** Markelle Kelly (2000). In Column 3, the Mean-square error (mean ± standard deviation) for the test data from models trained on the **Walker2d dataset** Lechner & Hasani (2020) is detailed. Column 4 and 5 show the AUC Performance of Models for Unseen Events in **Stance Classification** Leon Derczynski & Kochkina (2019). For all the datasets, every model is trained for 5 times with 5 different seeds.

| Model | Person Activity Test-Accuracy ↑ | Walker2d Test MSE ↓ | Sydneysiege AUC ↑ | Charliehebdo AUC ↑ |
|---|---|---|---|---|
| **Discrete Time Discrete Depth** | | | | |
| RNN-Decay Weinan (2017) | 78.74 ± 3.65 | 1.44 ± 0.01 | 0.62 ± 0.00 | 0.63 ± 0.01 |
| Bidirectional-RNN Schuster & Paliwal (1997) | 82.86 ± 1.17 | 1.09 ± 0.01 | 0.61 ± 0.00 | 0.64 ± 0.02 |
| GRU-D Che et al. (2016) | 82.52 ± 0.86 | 1.14 ± 0.01 | 0.63 ± 0.00 | **0.65 ± 0.01** |
| Phased-LSTM Neil et al. (2016) | 83.34 ± 0.59 | 1.10 ± 0.01 | 0.58 ± 0.01 | 0.60 ± 0.01 |
| **Continuous Time Discrete Depth** | | | | |
| CT-RNN ichi Funahashi & Nakamura (1993) | 82.32 ± 0.83 | 1.25 ± 0.03 | 0.56 ± 0.00 | 0.61 ± 0.01 |
| ODE-RNN Rubanova et al. (2019) | 75.03 ± 1.87 | 1.88 ± 0.05 | 0.55 ± 0.00 | 0.57 ± 0.03 |
| ODE-LSTM Lechner & Hasani (2020) | 83.77 ± 0.58 | 0.91 ± 0.02 | 0.56 ± 0.00 | 0.59 ± 0.00 |
| CT-GRU Mozer et al. (2017) | 83.93 ± 0.86 | 1.22 ± 0.01 | **0.63 ± 0.01** | 0.65 ± 0.02 |
| GRU-ODE De Brouwer et al. (2019) | 82.80 ± 0.61 | 1.08 ± 0.01 | 0.56 ± 0.00 | 0.61 ± 0.00 |
| CT-LSTM Lechner & Hasani (2020) | 83.42 ± 0.69 | 1.03 ± 0.02 | 0.62 ± 0.00 | 0.65 ± 0.01 |
| **Continuous Time Continuous Depth** | | | | |
| CDR-NDE Anumasa et al. (2023) | 87.54 ± 0.34 | 0.97 ± 0.04 | 0.57 ± 0.02 | 0.55 ± 0.01 |
| CDR-NDE-heat (Euler) | 88.24 ± 0.31 | 0.54 ± 0.01 | 0.62 ± 0.01 | 0.58 ± 0.02 |
| CDR-NDE-heat (Dopri5) | 88.60 ± 0.26 | 0.49 ± 0.01 | 0.62 ± 0.01 | 0.57 ± 0.01 |
| **Neural Wave - Single GRU** | 88.52 ± 0.34 | 0.49 ± 0.01 | 0.60 ± 0.01 | 0.63 ± 0.02 |
| **Neural Wave - Single MLP** | 90.58 ± 0.58 | **0.11 ± 0.01** | 0.59 ± 0.02 | 0.61 ± 0.01 |
| **Neural Wave - Double Gating** | **93.62 ± 0.38** | 0.16 ± 0.01 | 0.61 ± 0.01 | 0.62 ± 0.02 |
| **Neural Wave - MLP+GRU** | 92.06 ± 0.34 | 0.12 ± 0.01 | 0.60 ± 0.01 | 0.63 ± 0.04 |

## 5.3 SEPSIS PREDICTION USING PHYSIONET 2019 DATA

We analyze a dataset initially used in the PhysioNet 2019 challenge Reyna et al. (2019) Goldberger et al. (2000), focusing on sepsis prediction.

This dataset contains 40,335 sequences of variable lengths, documenting patient admissions in an intensive care unit (ICU), and includes five static features, such as patient age, as well as thirty-four dynamic features like Heart Rate, Blood pressure, etc. The measurements are taken at hourly intervals. A significant portion of the data is missing, with only 10.3% of the values being observed. Our analysis focuses on the initial 72 hours of a patiends stay, addressing the binary classification task of predicting sepsis development throughout their entire stay. We divided our data into a train, validation and test split of 70%, 15 % and 15% respectively. We compared Neural Wave's performance against GRU-ODE, GRU-D, ODE-RNN, Neural CDEKidger et al. (2020), CDR-NDE and GRU-$\Delta t$, a variant of GRU. that incorporates the time difference between observations as an additional input. We conduct experiments with various models considering the observational intensity. Observational intensity refers to the frequency of data observations, which can indicate the level of attention or concern, such as more frequent measurements for patients considered at higher risk ( more details mentioned in Section 3.5 and 3.6.). Table 2 illustrates the findings, where we use AUC for evaluation due to the datseds imbalance.

Table 2: Test AUC (mean ± standard deviation over five runs) for sepsis prediction on the PhysioNet.

| Model | Test AUC |
|---|---|
| GRU-ODE De Brouwer et al. (2019) | 0.852 ± 0.010 |
| GRU-$\Delta t$ | 0.878 ± 0.006 |
| GRU-D Che et al. (2016) | 0.871 ± 0.022 |
| ODE-RNN Rubanova et al. (2019) | 0.874 ± 0.016 |
| Neural CDE Kidger et al. (2020) | 0.880 ± 0.006 |
| CDR-NDE-heat Anumasa et al. (2023) | 0.880 ± 0.000 |
| **Neural Wave - Single GRU** | 0.885 ± 0.003 |
| **Neural Wave - Single MLP** | 0.885 ± 0.001 |
| **Neural Wave - Double Gating** | **0.890 ± 0.006** |
| **Neural Wave - MLP+GRU** | 0.889 ± 0.003 |

For the ODE-RNN, GRU-D, and GRU-$\Delta t$ models, observational intensity is given by appending an observed/not-observed mask to the input at each observation. The Neural CDE, GRU-ODE and Neural Wave model use a continuous, per-channel intensity as explained in Section 3.6 in Kidger

et al. (2020). The results demonstrate that the proposed model, **Neural Wave's - Double gating** provides the best performance, while other neural wave equation models are also equally competitive.

## 5.4 STANCE CLASSIFICATION

In practical scenarios, particularly on social media platforms like Twitter, tweets associated with a specific event are posted at varying times, and the intervals between these tweets are not uniform. We evaluate the models based on their ability to classify the stance of social media posts, specifically using the Twitter datasetLeon Derczynski & Kochkina (2019), which includes rumors associated with eight events. Each event comprises a collection of tweets labeled as Support, Query, Deny, or Comment. To create a sequence data point, we randomly selected 10 tweets and then sorted them by observation time in the ascending order. To evaluate our models, we considered an unseen event prediction setup, where the model performance is evaluated on the sequences from an unseen event. Here, the model is trained on the sequences formed from data from all the events except test event and tested on the sequences formed from the unseen test event. We selected two events: Sydneysiege and CharlieHebdo, as the test event for our experiments.

In Table 1, Columns 4 and 5 showcase the test AUC for unseen events for all models. We could not run the Neural CDE model due to lengthy epoch times, likely slowed by the cubic spline path construction, particularly as the input feature dimensions and sequence lengths increased. Despite the high dimensionality (316) of text embeddings in our stance classification data,neural wave equation models demonstrated robust performance. The stance classification task was intentionally chosen to check the performance of our model on a high dimensional discrete system. Even if our model does not beat some of the baselines, the **Neural Wave Model - MLP + GRU** remains competitive hence showing the robustness of our model.

## 5.5 ABLATION STUDIES

We conducted experiments with models considering a homogenous PDE with no source terms to understand the effect of source functions. The homogeneous neural wave equation model, without a source term, achieved a test accuracy of 51.73 % $\pm$ 0.16 on Person Activity, a test MSE of 0.99 $\pm$ 0.003 on Walker2D, and a test AUC of 0.857 $\pm$ 0.001 on Physionet Sepsis. We observe that the learnable source function helps to capture the dependencies present in the complex sequence modeling tasks more effectively and improves performance. From Table 1, we observe that the performance of the single GRU neural wave equation is lower than the rest. This is attributed to the fact that in the single GRU model, the source function captures nonlinear dependency between two neighboring hidden states whereas in the rest of the models, the nonlinear dependency is captured between all the four neighboring hidden states present in the FDM discretization of the wave equation. We study the memory consumption of our models on the PhysioNet data. Our models consume more memory, ranging from 1807MB to 2137MB, compared to the memory-efficient Neural CDE, which consumes up to 244MB. But, as we saw in the Person activity data, neural wave equations are an order of magnitude faster than neural CDE and are scalable to large sequence lengths. More details on time and memory complexity can be found in (A.7)).

## 6 CONCLUSION, LIMITATIONS AND FUTURE WORK

In this work, we propose neural wave equation, a sequence model based on the non-homogeneous wave equation. We show that a non-homogeneous wave equation with a learnable source function is a good fit for sequence modeling tasks involving irregularly sampled data. We establish that the analytical solution of a non-homogeneous wave equation presents a way to implicitly model denser connections between hidden states. We empirically demonstrate this by comparing our model against several baselines and outperforming them in several real-world data sets. Neural wave equations have reasonable computational speed, however this comes at the cost of memory consumption. Studying the benefits of using partial differential equations for sequence modeling with theoretical rigor and finding the correct balance between memory and speed is left for future work.

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

## A  APPENDIX / SUPPLEMENTAL MATERIAL

Most of the derivations regarding wave and heat equation can be found in more detail in Sobolev, Strauss and Evans (2010)

### A.1  SOLUTION OF WAVE EQUATION

$$u_{tt} = c^2 u_{xx} + F(t, x) \text{ where}$$
$$u(0, x) = f(x) \text{ and } u_t(0, x) = g(x)$$

We break it into two separate problems.

$$v_{tt} = c^2 v_{xx} \text{ where}$$
$$v(0, x) = f(x) \text{ and } v_t(0, x) = g(x)$$

and

$$w_{tt} = c^2 w_{xx} + F(t, x) \text{ where}$$
$$w(0, x) = 0 \text{ and } w_t(0, x) = 0$$

In such a case, the sum of the solutions of the above equations will give us the solution of the wave equation. To solve the first part Introduce new variables $\xi$ and $\eta$:

$$\xi = x - ct, \quad \eta = x + ct. \tag{12}$$

Then, the partial derivatives transform as follows:

$$\frac{\partial}{\partial x} = \frac{\partial \xi}{\partial x} \frac{\partial}{\partial \xi} + \frac{\partial \eta}{\partial x} \frac{\partial}{\partial \eta} = \frac{\partial}{\partial \xi} + \frac{\partial}{\partial \eta}, \tag{13}$$

$$\frac{\partial}{\partial t} = \frac{\partial \xi}{\partial t} \frac{\partial}{\partial \xi} + \frac{\partial \eta}{\partial t} \frac{\partial}{\partial \eta} = -c\frac{\partial}{\partial \xi} + c\frac{\partial}{\partial \eta}. \tag{14}$$

The second derivatives are:

$$\frac{\partial^2}{\partial x^2} = \left( \frac{\partial}{\partial \xi} + \frac{\partial}{\partial \eta} \right)^2 = \frac{\partial^2}{\partial \xi^2} + 2\frac{\partial^2}{\partial \xi \partial \eta} + \frac{\partial^2}{\partial \eta^2}, \tag{15}$$

$$\frac{\partial^2}{\partial t^2} = \left( -c\frac{\partial}{\partial \xi} + c\frac{\partial}{\partial \eta} \right)^2 = c^2 \left( \frac{\partial^2}{\partial \xi^2} - 2\frac{\partial^2}{\partial \xi \partial \eta} + \frac{\partial^2}{\partial \eta^2} \right). \tag{16}$$

**Substitute into the Wave Equation**

Substitute these into the wave equation:

$$c^2 \left( \frac{\partial^2 v}{\partial \xi^2} - 2\frac{\partial^2 v}{\partial \xi \partial \eta} + \frac{\partial^2 v}{\partial \eta^2} \right) = c^2 \left( \frac{\partial^2 v}{\partial \xi^2} + 2\frac{\partial^2 v}{\partial \xi \partial \eta} + \frac{\partial^2 v}{\partial \eta^2} \right). \tag{17}$$

Simplify this to:

$$0 = 4c^2 \frac{\partial^2 v}{\partial \xi \partial \eta}. \tag{18}$$

This implies:

$$\frac{\partial^2 v}{\partial \xi \partial \eta} = 0. \tag{19}$$

The solution of the above equation can be found by integrating twice. So, we know that the solution will be of the form

$$v(x, t) = A(x - ct) + B(x + ct) \tag{20}$$

Now, $A(x) = \frac{1}{2}f(x) - \frac{1}{2c} \int_0^x g(s)ds$ and $B(x) = \frac{1}{2}f(x) + \frac{1}{2c} \int_0^x g(s)ds$ solves the first part of the wave equation. The solution then can be written down as

$$v(x, t) = \frac{f(x - ct) + f(x + ct)}{2} + \frac{1}{2c} \int_{x-ct}^{x+ct} g(s)ds \tag{21}$$

To solve the second part, let us consider another initial value formulation of the wave equation.

$$r_{tt} = c^2 r_{xx} \text{ where } r(\tau, x; \tau) = 0 \text{ and } r_t(\tau, x; \tau) = F(\tau, x)$$

In this case, $w(t, x) = \int_0^t r(\tau, x; \tau) d\tau$ solves the second part of the wave equation. Using the Leibnitz rule for differentiation under integral sign, we can write,

$$w_t = r(t, x; t) + \int_0^t r_t(t, x; \tau) d\tau = \int_0^t r_t(t, x; \tau) d\tau$$

$$w_{tt} = r_t(x, t; t) + \int_0^t r_{tt}(t, x; \tau) d\tau = F(t, x) + \int_0^t r_{tt}(t, x; \tau) d\tau$$

and we have

$$w_{xx} = \int_0^t r_{xx}(t, x; \tau) d\tau = \tfrac{1}{c^2} \int_0^t r_{tt}(t, x; \tau) d\tau$$

putting the values of $w_{xx}$ and $w_{tt}$ in the equation, we get

$$w_{tt} - c^2 w_{xx} = F(t, x) \tag{22}$$

By D'Alemberds formula, the solution of this initial value problem is

$$r(t, x; \tau) = \frac{1}{2c} \int_{x-c(t-\tau)}^{x+c(t-\tau)} F(\tau, \eta) d\eta \tag{23}$$

and

$$w(t, x) = \frac{1}{2c} \int_0^t \int_{x-c(t-\tau)}^{x+c(t-\tau)} F(\tau, \eta) d\eta \tag{24}$$

Adding the solutions of both the parts, we get the solutions of wave equations as

$$u(t, x) = \frac{f(x - ct) + f(x + ct)}{2} + \frac{1}{2c} \int_{x-ct}^{x+ct} g(s) ds + \frac{1}{2c} \int_0^t \int_{x-c(t-\tau)}^{x+c(t-\tau)} F(\tau, \eta) d\eta \tag{25}$$

## A.2 FDM DISCRETIZATION OF WAVE EQUATION

$$\frac{\partial^2 h_{t,d}}{\partial d^2} - \frac{\partial^2 h_{t,d}}{\partial t^2} = 0 \tag{26}$$

Discretization of the 1D Wave equation is as follows:

$$\frac{(h_{(t,d+\Delta d)} - 2h_{(t,t)} + h_{t,d-\Delta d})}{\Delta_d^2} - \frac{(h_{t-\Delta t,d} - 2h_{t,d} + h_{t+\Delta t,d})}{\Delta_t^2} = 0 \tag{27}$$

$$\frac{(h_{(t,d+\Delta d)})}{\Delta_d^2} = \frac{(2 * h_{t,d} - h_{(t,d-\Delta d)})}{\Delta_d^2} - \frac{(h_{t-\Delta t,d} - 2h_{t,d} + h_{t+\Delta t,d})}{\Delta_t^2} \tag{28}$$

$$h_{t,d+\Delta_d} = 2h_{t,d} - h_{t,d-\Delta_d} + \frac{\Delta_d^2}{\Delta_t^2}[h_{t+\Delta t,d} - 2h_{t,d} + h_{t-\Delta t,d}] \tag{29}$$

## A.3 SOLUTION OF HEAT EQUATION

$$u_t = ku_{xx} + Q(x, t) \text{ where } u(0, t) = 0, u(L, t) = 0, u(x, 0) = f(x)$$

Using separation of variables (assuming that the solution is of the form $u(x, t) = X(x)T(t)$) leads to an eigenvalue problem

$$\phi'' + \lambda\phi = 0, \phi(0) = 0, \phi(L) = 0$$

The eigenfunctions and eigenvalues are given by

$$\phi_n(x) = sin\frac{n\pi x}{L}, \lambda_n = (\frac{n\pi}{L})^2$$

Leds assume that the solution is off the form

$$u(x,t) = \sum_{n=11}^{\infty} a_n(t)\phi_n(x) \tag{30}$$

We write

$$f(x) = u(x,0) = \sum_{1}^{\infty} a_n(0)\phi_n(x) \tag{31}$$

$$Q(x,t) = \sum_{1}^{\infty} q_n(t)\phi_n(x) \tag{32}$$

The coefficients of the above equations are solved using the Fourier series. We expand

$$u_t(x,t) = \sum_{1}^{\infty} a'_n(t)\phi_n(x), \, u_{xx}(x,t) = -\sum_{1}^{\infty} a_n(t)\lambda_n\phi_n(x)$$

Inserting into the heat equation we get,

$$u_t = ku_{xx} + Q(x,t)$$
$$\sum_{1}^{\infty} a'_n(t)\phi_n(x) = -k\sum_{1}^{\infty} a_n(t)\lambda_n\phi_n(x) + \sum_{1}^{\infty} q_n(t)\phi_n(x)$$

$$a'_n(t) + k\lambda_n a_n(t) = q_n(t) \tag{33}$$

Solving the above ODE, we get

$$a_n(t)\exp^{k\lambda_n t} = a_n(0) + \int_0^t q_n(\tau)\exp^{k\lambda_n t} \tag{34}$$

$$a_n(t) = a_n(0)e^{-k\lambda_n t} + \int_0^t q_n(\tau)e^{-k\lambda_n(t-\tau)}d\tau \tag{35}$$

So, we write the solution as follows -

$$u(x,t) = \sum_{1}^{\infty} a_n(t)\phi_n(x) = \sum_{1}^{\infty}[a_n(0)e^{-k\lambda_n t} + \int_0^t q_n(\tau)e^{-k\lambda_n(t-\tau)}d\tau]\phi_n(x) \tag{36}$$

## A.4 SOLVING A 2ND ORDER EQUATION AS A SYSTEM OF 1ST ORDER EQUATIONS

In this case, since the entire sequence is fed at once as input, we know the values of $y_{xx}$ So, we can write the 2nd-order wave equation as a system of 1st-order odes.

$$Y = \begin{bmatrix} y \\ y_t \end{bmatrix} \tag{37}$$

$$Y_t = \begin{bmatrix} 0 & 1 \\ 0 & 0 \end{bmatrix} Y + \begin{bmatrix} 0 \\ c^2 y_{xx} + F(x,t) \end{bmatrix} \tag{38}$$

## A.5 EXAMPLE OF SOLUTION PROBLEM IN SOLVER

$$y_{tt} = y_{xx}$$
$$y(0,t) = y(L,t) = 0, y(x,0) = f(x), y_t(x,0) = g(x)$$

By D'Alembert's formula, we know the solution is

$$y(x,t) = \frac{f(x-ct) + f(x+ct)}{2} + \frac{1}{2c}\int_{x-ct}^{x+ct} g(s)ds \tag{39}$$

In the question, $f(x) = sin(\pi x)$ and $g(x) = sin(\pi x)$ So, the solution is

$$y(x,t) = \frac{sin\pi(x-t) + sin\pi(x+t)}{2} + \frac{1}{2}\int_{x-t}^{x+t} sin(\pi x)dx = sin(\pi x)[cos(\pi t) + \frac{sin(\pi t)}{\pi}] \tag{40}$$

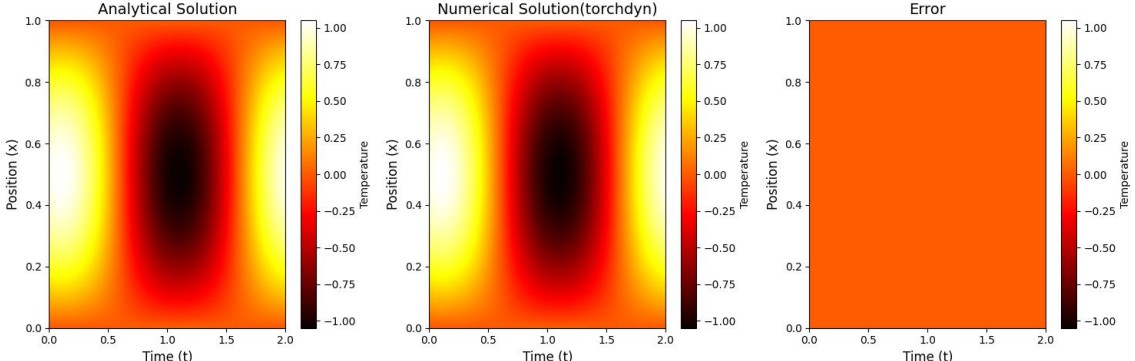

Figure 4: The leftmost figure is analytical solution over a grid,the middle figure is numerical solution, the last figure is the error between the analytical and numerical solution. We notice that the error between analytical and numerical solution is of the order $1e-3$ which was used as the relative error tolerance.

### A.6 FAST IMPLEMENTATION

ODESolvers are not suited to handle a sequence of vector data. The initial value condition in neural wave equation, $\mathbf{h}_{:,0}$ is an $N \times M$ matrix where N is the sequence length and M is the number of input features. While training with batches, $B$ being the batch size, the initial value condition becomes a $B \times N \times M$ matrix. To efficiently utilize GPUs during training, we collapse the batch and the sequence length into a single dimension resulting in a $(BN) \times M$ matrix. During training, the input sequence is a tuple of $(batch size X sequence length X input features)$. However, neural ODE solvers can't handle such data directly. One way to overcome this problem is to loop over batch. However, it is not GPU efficient. The elegant solution is to collapse batch size and sequence length into a single dimension and convert the 3d input array into a 2d array. The input matrix looks like -

$$\left[ X_{t_1}^1 \, X_{t_1}^2 .. X_{t_1}^b \, X_{t_2}^1 \, X_{t_2}^2 .. X_{t_2}^b .. X_{t_n}^1 \, X_{t_n}^2 .. X_{t_n}^b \right]^{\text{T}}$$

$X_{t_j}^i$ is the input vector corresponding to ith batch at time sequence j. We append this matrix at the start and at the end by repeating the first and last time sequence of every batch. For example if we assume the batch size to be 3, and sequence length to be 3, we will have the following matrix

$$\left[ X_{t_1}^1 \, X_{t_1}^2 \, X_{t_1}^3 \, X_{t_1}^1 \, X_{t_1}^2 \, X_{t_1}^3 \, X_{t_2}^1 \, X_{t_2}^2 \, X_{t_2}^3 \, X_{t_3}^1 \, X_{t_3}^2 \, X_{t_3}^3 \, X_{t_3}^1 \, X_{t_3}^2 \, X_{t_3}^3 \right]^{\text{T}}$$

Let this matrix be called $h$. Note that shifting this matrix lets us calculate the finite difference terms easily. For example

$$h[2b:] - 2h[b:-b] + h[:-2b] = \begin{bmatrix} X_{t_2}^1 \\ X_{t_2}^2 \\ X_{t_2}^3 \\ X_{t_3}^1 \\ X_{t_3}^2 \\ X_{t_3}^3 \\ X_{t_3}^1 \\ X_{t_3}^2 \\ X_{t_3}^3 \end{bmatrix} - 2 \begin{bmatrix} X_{t_1}^1 \\ X_{t_1}^2 \\ X_{t_1}^3 \\ X_{t_2}^1 \\ X_{t_2}^2 \\ X_{t_2}^3 \\ X_{t_3}^1 \\ X_{t_3}^2 \\ X_{t_3}^3 \end{bmatrix} + \begin{bmatrix} X_{t_1}^1 \\ X_{t_1}^2 \\ X_{t_1}^3 \\ X_{t_1}^1 \\ X_{t_1}^2 \\ X_{t_1}^3 \\ X_{t_2}^1 \\ X_{t_2}^2 \\ X_{t_2}^3 \end{bmatrix} \tag{41}$$

We can then append the necessary boundary conditions to the calculated matrix and pass it into the solver again. However, the shifting of matrix for calculating finite difference and source terms must be done carefully so that the batches does not mix amongst themselves.

## A.7 COMPUTATIONAL COMPLEXITY

We also analysed the computational complexity of neural wave equation with other baselines. Our implementation indeed uses more memory but is much faster compared to the exisiting methods.

| Model | Memory (in MB) | Speed (epoch/s) |
|---|---|---|
| **CTRNN** | 321 | 259 |
| **ODE-LSTM** | 348 | 82.39 |
| **CTGRU** | 662 | 11.45 |
| **GRUODE** | 161 | 11.93 |
| **Average** | **373** | **90.445** |
| **BIRNN** | 162 | 30.66 |
| **GRUD** | 48 | 13.22 |
| **PHASED** | 38 | 9.4 |
| **Average** | **82.67** | **17.76** |
| **Neural Wave** | **1972** | **8.66** |
| **Neural Wave with checkpointing** | **400** | **12** |

Table 3: Computational Complexity Analysis of Neural Wave Equation with other models

It is important to emphasize that the observed speed-memory tradeoff arises from our implementation technique rather than being an inherent property of the model. Specifically, in most RNN variants with ODE solvers, it is necessary to loop over the sequence dimension because ODE solvers typically cannot handle 3-dimensional data directly.To address this limitation, we collapsed the batch and sequence dimensions into a single dimension. This approach enabled us to utilize the GPU more efficiently, significantly improving speed by eliminating the need for looping. However, this optimization leads to a higher peak memory allocation, as the entire collapsed batch-sequence matrix must fit in memory during computation. If memory constraints arise, gradient checkpointing can reduce the neural wave equation's memory usage by 80%, albeit with a 50% increase in training time due to recomputation overhead.

## A.8 LOSS FUNCTION AND TRAINING

The loss function used to train the model depends on the problem. We use a cross-entropy loss for classification problems and mean-squared error for regression problems. The loss is a function of the MLP parameters $\theta_{MLP} = (\theta_{pre}, \theta_{post})$ and source function parameters $\theta_s$. Since we use the neural ODE framework during training, we prefer the adjoint sensitivity method over the traditional backpropagation as it offers memory efficiency Chen et al. (2018)Choon et al. (2019). Even though we mention ODESolvers to solve the wave equation, there are two practical problems one may face during training. The ODESolvers normally have an ODEFunc argument which is $\frac{\partial h(t,d)}{\partial d}$ in the wave equation. However, the wave equation is a 2nd-order PDE, and hence $\frac{\partial h(t,d)}{\partial d}$ is not known. It is required to convert the wave equation to a system of linear 1st-order equations. We define $H(d) = [h_{t,d}, \frac{\partial h_{t,d}}{\partial d}]$, and consider a first-order system $H(d) = G(\theta_s, H(d), d)$. Note that here, G is a function of both $h_{t,d}$ and $\frac{\partial h_{t,d}}{\partial d}$. Following this, we define the adjoint state as $a(d) = \frac{dL}{dH(d)}$, and an ODE which satisfies,

$$\frac{dH}{dd} = -a(d)\frac{\partial G(\theta_s, H(d), d)}{\partial H(d)} \tag{42}$$

we find $H(d)$ by making an extra call to the ODESolver with $\frac{dL}{dH(D)}$ as the initial condition.

## A.9 COMPARISON

### D'ALEMBERT SOLUTION OF THE WAVE EQUATION

The wave equation with a source term is given by

$$h(t, d) = \frac{1}{2}(f(t + cd) + f(t - cd)) + \frac{1}{2c}\int_0^d \int_{t-c(t-\tau)}^{t+c(t-\tau)} F(s, \tau)dsd\tau \tag{43}$$

where $h(t, 0) = f(t)$ and $F(.)$ is the source term.

COMPARISON WITH RNN TYPE ARCHITECTURES

In a normal RNN architecture, the evolution of the hidden state dynamics is as follows:

$$h_{t,d} = F(W_t h_{t-1,d} + W_{d-1} h_{t,d-1}) \tag{44}$$

So, the hidden state at point $(t, d)$ depends only on $h_{t-1,d}$ and $h_{t,d-1}$. In the wave equation, the presence of the integral over the source term from 0 to d ensures that each $h_{t,d}$ is modeled as a function of several hidden states. The trainable parameter c determines the number of the hidden states with depth $< d$ that contributes to the evolution of $h_{t,d}$.

COMPARISON WITH HEAT EQUATION

It has been shown that heat equation can also be used to model sequence data. The discretization of the heat equation is:

$$h(t, d + \Delta_d) = \frac{\Delta_d}{\Delta_t}[h_{(t-\Delta_t, d)} - 2h_{(t,d)} + h_{(t+\Delta_t, d]} + h_{(t,d)} \tag{45}$$

Straight up comparing with the discretization of the wave equation, we notice that the $h_{(t, d - \delta_d)}$ term is absent in the heat equation. The analytical solution of the heat equation is:

$$h(t, d) = \sum_{n=1}^{\infty} (a_n(0) \exp^{(-k\lambda_n d)} + \int_0^d q_n(\tau) \exp^{(-k\lambda_n(t-\tau)} d\tau) \phi_n(t) \tag{46}$$

where $\phi_n = sin\frac{(n\pi t)}{T}, q_n = \int_0^T F(t, d)\phi_n(t)dt$.

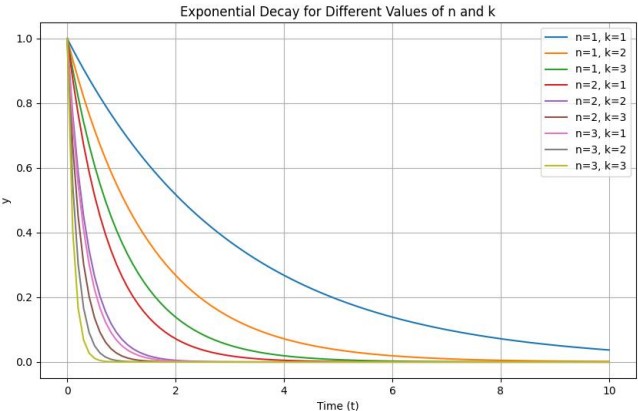

Figure 5: The rate of exponential decay in heat equation corresponding to different depth and heat diffusivity.

The presence of the negative exponential term in the solution of the heat equation means that the effect of the hidden states located at lower depths is diminished while calculating the hidden states located at higher depths. The wave equation does not have this problem. In the case of the depth of the model being shallow, heat and wave equations show similar performance.

A.10    NEURAL WAVE EQUATION ALGORITHM

The algorithm for neural wave equation is provided. The architecture diagram of the encoder-decoder version is also provided.

---

**Algorithm 1** Neural Wave Equation

---

**function** PRE-NN(input data)

    Compute the initial condition using a neural network

    $h(t, 0) \leftarrow$ MLP(input data)

    **return** $h(t, 0)$

**end function**

**function** ODEFUNC(source function, current state)

    Compute the second-order time derivative:

    second derivative $\leftarrow$ Finite Difference Formula + source function

    Store the current state as it will be required to calculate the FDM for next state.

    **return** second derivative

**end function**

**function** NEURALWAVE(initial condition)

    Initialize a second-order ODE solver

    Solve the wave equation using the solver

    **return** the solution at final time $D$

**end function**

**function** POST-NN(solution at $D$)

    Apply a neural network for post-processing

    output $\leftarrow$ MLP(solution at $D$)

    **return** output

**end function**

---

## A.11    ADAPTIVE STEP SIZE SOLVERS

$$y'(t) = f(t, y(t)), y(a) = y_a \tag{47}$$

The exact solution at nth point is $y_n$ and the numerical approximation is $\bar{y_n}$. Approximation of $y_n$ using RK-4 method yields

$$y_n^{RK4} = \bar{y_n} + O(\Delta_t^5) \tag{48}$$

while RK-5 yields,

$$y_n^{RK5} = \bar{y_n} + O(\Delta_t^6) \tag{49}$$

$$\epsilon = |y_n^{RK5} - y_n^{RK4}| = O(\Delta_t^5) \tag{50}$$

Given a relative error tolerance, we can calculate the required step size $\Delta_\tau$ by solving:

$$\frac{\epsilon}{tol} = \frac{\Delta_t^5}{\Delta_\tau^5} \tag{51}$$

$$\Delta_\tau = (\frac{tol}{\epsilon})^{\frac{1}{5}} \Delta_t \tag{52}$$

## A.12    BOUNDARY CONDITIONS

For numerically solving a partial differential equation or an ODE, we need boundary conditions or initial value conditions. Since, we are using a numerical solver, we also need a list of initial value conditions. Here, we aim to discuss in detail how the initial value conditions can be initialized in case of the neural wave equation. First let us take a look at the update equation again. For simplification, we look at the discretization equation of homogenous wave equation.

$$h_{t,d+\Delta_d} = 2h_{t,d} - h_{t,d-\Delta_d} + \frac{\Delta_d^2}{\Delta_t^2} c^2 [h_{t+\Delta_t,d} - 2h_{t,d} + h_{t-\Delta_t,d}]$$

We have a sequence of data at different time points (t) to begin with. $h_{t,0}$ corresponds to these data. (please note that instead of the raw data, we often pass them through a MLP to get the initial values at $h_{t,0}$. This is mainly to reduce or increase the dimension of the data.) Now, let us see how $h_{0,0+\Delta_d}$ gets calculated. The update equation will read as follows

$$h_{t,0+\Delta_d} = 2h_{t,0} - h_{t,0-\Delta_d} + \frac{\Delta_d^2}{\Delta_t^2} c^2 [h_{t+\Delta_t,0} - 2h_{t,0} + h_{t-\Delta_t,0}]$$

The $h_{t,0-\Delta_d}$ value is not available to us and we can use a value of our choice as the boundary condition. Please note that this is same as specifying the partial derivative of h wrt t' at point 0. If

we write our own custom implementation of a solver, we can follow one of the common scheme for initializing boundary values like dirichlet, neumann or robins. However, in our implementation, we used the torchdyn solver which takes the derivative at time 0 as 0. Now, we look at the second set of points where we need boundary conditions. Let's take a look at the update rule for $h_{0,d+\Delta_d}$

$$h_{0,d+\Delta_d} = 2h_{0,d} - h_{0,d-\Delta_d} + \frac{\Delta_d^2}{\Delta_t^2}c^2[h_{0+\Delta_t,d} - 2h_{t,d} + h_{0-\Delta_t,d}]$$

Here, $h_{0-\Delta_t,d}$ is again not known to us and we need to tackle it just like the above case. Again note that this is same as specifying the derivative wrt t at 0 and we use 0 in our implementation. However, one can come up with custom boundary condition according to the problem in hand.

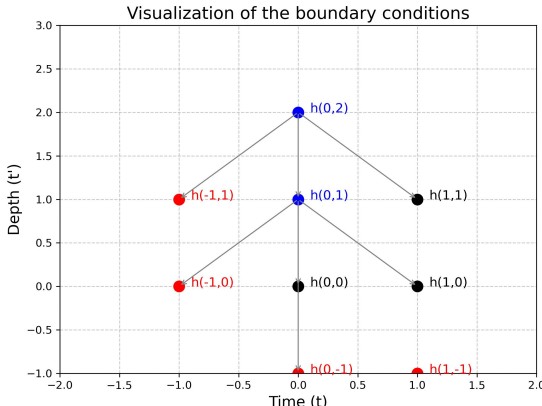

Figure 6: Visualization of the boundary conditions for solving the PDE. The red cells represent the **boundary conditions** that must be defined to compute the values at the blue cells ($h_{0,1}$ and $h_{0,2}$). These boundary conditions correspond to unknown values, such as $h_{-1,1}$, $h_{-1,0}$, and $h_{0,-1}$, which are set based on the problem or solver design (e.g., Dirichlet or Neumann boundary conditions). The black cells denote the values that are either **given** (e.g., initial conditions) or **computed** as part of the numerical solution process.

A.13 EXPERIMENTS

Table 4 outlines the numerical methods selected for each model. The Neural Wave model employs the Dopri5/Tsit5 method, setting the absolute and relative tolerance levels to $1e^{-3}$. A scheduled learning rate decay strategy is implemented, with a decay coefficient $\gamma = 0.1$, activated at the 100th epoch.

Table 4: ODE solvers used for different RNODE models. For the Neural Wave model using Dopri5, the absolute and relative tolerance values are $1e^{-3}$ and $1e^{-3}$ respectively.

| Model | ODE-Solver | Time-step Ratio |
|---|---|---|
| CT-RNN ichi Funahashi & Nakamura (1993) | 4-th order Runge-Kutta | $1/3$ |
| ODE-RNN Rubanova et al. (2019) | 4-th order Runge-Kutta | $1/3$ |
| GRU-ODEDe Brouwer et al. (2019) | Explicit Euler | $1/4$ |
| ODE-LSTM Lechner & Hasani (2020) | Explicit Euler | $1/4$ |
| Neural-CDE Kidger et al. (2020) | Dopri5 | - |
| CDR-NDE Anumasa et al. (2023) | Explicit Euler | $1/2$ |
| CDR-NDE-heat Anumasa et al. (2023) | Dopri5 | - |
| Neural Wave | Dopri5/Tsit5 | - |

WALKER V2 KINEMATICS

The output is a 17-dimensional vector at each time point, we visualize the comparison between the ground truth and the predicted values across several randomly selected time points over test samples.

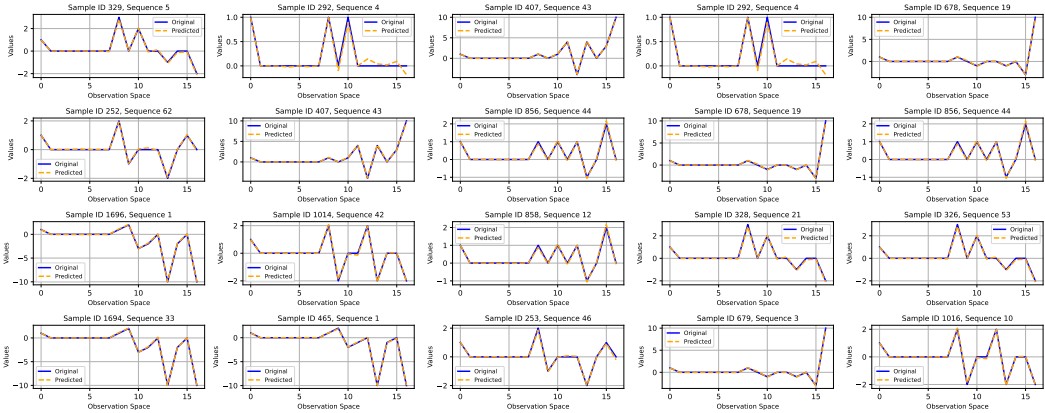

Figure 7: Comparison between ground truth and predicted position of the observation space of the walker2d kinematics model.

STANCE CLASSIFICATION

Table 5: Test AUC Performance for Models on Seen Events

| Model | AUC (Sydneysiege event) | AUC (Charliehebdo event) |
|---|---|---|
| CT-RNN | 0.57 ± 0.00 | 0.63 ± 0.01 |
| ODE-RNN | 0.55 ± 0.01 | 0.59 ± 0.02 |
| ODE-LSTM | 0.56 ± 0.01 | 0.61 ± 0.01 |
| CT-GRU | 0.64 ± 0.01 | 0.67 ± 0.01 |
| RNN-Decay | 0.63 ± 0.01 | 0.67 ± 0.02 |
| Bidirectional-RNN | 0.62 ± 0.01 | 0.67 ± 0.00 |
| GRU-D | 0.64 ± 0.01 | 0.69 ± 0.01 |
| Phased-LSTM | 0.61 ± 0.01 | 0.64 ± 0.01 |
| GRU-ODE | 0.56 ± 0.00 | 0.63 ± 0.01 |
| CT-LSTM | 0.64 ± 0.01 | 0.66 ± 0.04 |
| Augmented-LSTM | 0.64 ± 0.01 | 0.68 ± 0.00 |
| CDR-NDE | 0.57 ± 0.01 | 0.60 ± 0.01 |
| CDR-NDE-heat(Euler) | 0.64 ± 0.01 | 0.66 ± 0.01 |
| CDR-NDE-heat(Dopri5) | 0.63 ± 0.01 | 0.65 ± 0.01 |
| **Neural Wave - Single GRU** | 0.59 ± 0.01 | 0.62 ± 0.01 |
| **Neural Wave - Single MLP** | 0.60 ± 0.02 | 0.65 ± 0.01 |
| **Neural Wave - Double Gating** | 0.60 ± 0.01 | 0.60 ± 0.01 |
| **Neural Wave - MLP+GRU** | 0.61 ± 0.02 | 0.64 ± 0.01 |

SENSITIVITY OF WAVE SPEED PARAMETER C

We conducted a small study on the homogeneous model excluding the neural network, as it likely compensates for suboptimal initial values of c on the Walker dataset.

This allowed us to isolate the impact of c without the corrective influence of a trainable network. The results (A.13) show that test loss varies with c. For the model where c is trainable, the MSE is 0.99. This suggests that the flexibility of a trainable c mitigates sensitivity, improving overall robustness.

| $c$ | Test Loss |
|---|---|
| 0.1 | 2.728 |
| 0.5 | 2.429 |
| 1 | 1.281 |
| 3 | 2.329 |
| 5 | 6.243 |

