# OpenReview forum: "Neural Wave Equation for Irregularly Sampled Sequence Data"
_ICLR.cc/2025/Conference — ICLR 2025 Poster_

### Official Review · Reviewer_Fbp1 · 2024-11-01

**Soundness:** 4
**Presentation:** 3
**Contribution:** 3
**Rating:** 6
**Confidence:** 3

**Summary:**

This paper addresses the limitation of fixed architectural depth in modeling irregularly sampled sequence data. A neural wave equation method is proposed to 1) continuously model the hidden states across time and depth and 2) allow denser connections across hidden states. Empirical results show improved performance compared to baseline methods.

**Strengths:**

1. The problem of fixed assumption in architectural depth is important.
2. The method of using the wave equation to continuously model the time and depth is creative. Developing the neural network based on the analytical solution to the wave equation allows more dependencies in hidden states.
3. Experiments were performed on different scenarios to show the effectiveness of the proposed method.

**Weaknesses:**

1. The introduction and related work need to be better structured. For instance, what is the concept of hidden state depth in real-world problems? Could the authors elaborate on why PDE-based models are a better model for continuous modeling on depth? What is the benefit of the wave equation over the other PDEs? The authors could provide more details in both sections for a smoother transition from the problem setting to the proposed method.
2. Section 3 and 4 could be better presented. For example, instead of presenting Figure 2, the authors could provide a comparison between ODE-RNN and the proposed method in terms of how they model in depth. In fact, Figure 2 should be in an ablation study and needs a better explanation of how the depth affects the three comparison models. Also, using $t’$ for depth is confusing given that $t$ is used for time. For Figure 3, the authors should explain the blue arrows with the description in Section 4.2.
3. The experiments should be performed and analyzed in more detail. For each experiment, could the authors highlight the purpose, such as how the depth of hidden states or the number of missing data affects the model performance? What is the depth of each group of data? Could the authors group the comparison baselines as the categories in Figure 1? Table 1 could be separated by the experiments. Also, it would be better to have visualizations about the predicted sequences vs the ground truth to better understand the improvement in metrics.

**Questions:**

Please check the questions listed in the Weaknesses section.

---

> ### Author Response · Authors · 2024-11-22
> **Response to reviewer Fbp1**
>
> Thank you for taking the time to review our submission and providing valuable feedback. We greatly appreciate your insightful comments and suggestions, which have helped us improve the quality and clarity of our work.
>
> **The introduction and related work need to be better structured. For instance, what is the concept of hidden state depth in real-world problems? Could the authors elaborate on why PDE-based models are a better model for continuous modeling on depth? What is the benefit of the wave equation over the other PDEs? The authors could provide more details in both sections for a smoother transition from the problem setting to the proposed method.**
>
> We have updated the introduction and the related work in the revised submission of our work.
>
> We have provided more justifications on why PDE based models are better for continuous modeling on depth in Section 1 (line 66-69). The benefit of the wave equation over other PDEs can be found in section 1 (line 78-84).
>
> We chose the wave equation because its FDM discretization reveals that the evolution of hidden states depends on their neighbors, and it includes an additional term in depth not present in the heat equation. (Section 1 : line 78-84) ). This additional term has a significant effect on the properties of the wave equation as it enables a better flow of information compared to the heat equation. We have provided a more detailed discussion and analytical  comparison in section 4.3.
>
> The analytical solution of non-homogeneous wave equations allows for a deeper study of their theoretical properties, providing a key motivation for this work: to investigate, from an analytical perspective, why PDEs are effective in modeling sequence data which is not the case for a large number of non-homogenous PDEs.
>
> **Section 3 and 4 could be better presented. For example, instead of presenting Figure 2, the authors could provide a comparison between ODE-RNN and the proposed method in terms of how they model in depth. In fact, Figure 2 should be in an ablation study and needs a better explanation of how the depth affects the three comparison models. Also, using t′ for depth is confusing given that t is used for time. For Figure 3, the authors should explain the blue arrows with the description in Section 4.2.**
>
> In ODE-RNN, depth is modeled explicitly by stacking multiple ODE-RNN cells one after the other, similar to traditional RNN architectures. Each ODE-RNN cell consists of two primary components: an RNN cell, which performs discrete updates at each time step t, and a Neural ODE, which governs the continuous evolution of the hidden state between discrete observations. The depth in ODE-RNN refers to the number of such stacked cells, where each cell transforms the hidden state sequentially.
>
> In Neural Wave Equation, there is no explicit concept of depth as seen in traditional neural networks with stacked layers. Instead, the transformations of the hidden states are governed by a partial differential equation, and the numerical solver performs several transformations of the hidden states simulating depth. This continuous perspective eliminates the need for discrete, layered depth. The number of transformations by the numerical solver depend on the input and automates depth selection in neural wave equations.
>
> We have updated section 3.3 to provide a better understanding of ODE-RNN and comparing it with neural wave equations with the help of Figure 1.
>
> We need to include Figure 2 in Section 3  to highlight the importance of automating  depth selection and consequently provide a strong motivation for the neural wave equation. .
>
> We have changed t’ to d since, it denotes the depth direction for better readability.
>
> The diagram illustrates the structured grid used for solving a PDE using a finite difference scheme.The black arrows represent the grid cells that contribute to solving for $h_{t,d+\Delta_{d}}$ at the current time step. These cells are part of the numerical stencil, which defines the dependencies used in the FDM. These arrows indicate the grid cells that contribute to the computation of the source function at the current grid point.

---

> > ### Author Response · Authors · 2024-11-22
> > **Response to reviewer Fbp1 (continued)..**
> >
> > **The experiments should be performed and analyzed in more detail. For each experiment, could the authors highlight the purpose, such as how the depth of hidden states or the number of missing data affects the model performance? What is the depth of each group of data? Could the authors group the comparison baselines as the categories in Figure 1? Table 1 could be separated by the experiments. Also, it would be better to have visualizations about the predicted sequences vs the ground truth to better understand the improvement in metrics.**
> >
> > The person’s activity,walker v2 kinematics and physionet dataset are standard datasets for evaluation of continuous time dynamics models [1,2,3]. The reasons these datasets are chosen is for evaluating irregular sampled time series data arising from  continuous time dynamics.  Stance classification was purposefully chosen to make sure that our model also performs competitively on a dataset which is high dimensional and not generated from a continuous process.
> >
> > In RNN or ODE-RNN variants, we would have to stack layers of RNN/ODE-RNN cells on top of each other to simulate depth.Neural Wave Equation has no concept of layer wise depth like we see in case of other neural networks. The closest analog we have to depth in case of neural wave equation is the number of steps taken by the solver which is decided automatically by the adaptive step size solver leveraging the local error (see Appendix A.10) unless a tolerance criteria is met.  We have mentioned in section 5.1 that the average number of steps (or function calls) by the solver is 26 in case of Walker 2d Kinematics and 32 on Person’s activity dataset in Experiment section under subsection Person’s activity. This variation in the number of function calls across datasets highlights the neural wave equation's ability to implicitly model depth in accordance with the complexity of the data, adapting dynamically to the underlying patterns and challenges of different tasks.
> >
> > We have grouped the baselines according to categories in figure 1 as per your suggestion. We kept the experiments together because of space constraints.
> >
> > We have added visualization plots for walker in Appendix A.13.
> >
> > 1.M Lechner, R Hasani,Mixed-Memory RNNs for Learning Long-term Dependencies in Irregularly Sampled Time Series, NeurIPS 2022 Memory in Artificial and Real Intelligence workshop
> >
> > 2.Y Rubanova, RTQ Chen, DK Duvenaud,Latent ordinary differential equations for irregularly-sampled time series,Advances in neural information processing systems 32
> >
> > 3.S Anumasa, G Gunapati, PK Srijith, Continuous Depth Recurrent Neural Differential Equations,Joint European Conference on Machine Learning and Knowledge Discovery in Databases
> >
> > We hope that our revisions have resolved the issues you raised. If you find our responses satisfactory and the changes adequate, we kindly ask you to consider reflecting this in your overall evaluation.

---

> > > ### Comment · Reviewer_Fbp1 · 2024-11-24
> > > **Thanks for reply**
> > >
> > > Thanks for the authors' response and efforts to improve the presentation of the work. I will raise my rating.

---

### Official Review · Reviewer_HEKw · 2024-11-02

**Soundness:** 4
**Presentation:** 2
**Contribution:** 4
**Rating:** 8
**Confidence:** 4

**Summary:**

This paper introduces a novel architecture for latent space modeling and prediction using the Wave equation. Motivated by the homogeneous and non-homogeneous wave equations, the authors proposed neural wave equation to model the hidden layer transformations. In the experiments, the authors tested and showed superior performances on person activity recognition, walker2d-v2, sepsis, and social media posts datasets.

**Strengths:**

This paper proposed a very innovative idea which utilizes the wave equation to model the latent space. From the reviewer's perspective, this is good since this kind of structure could model physical/natural phenomenon(s) better. And the model is still generalizable from the non-homogeneous formulation with F(z,t).

Experimentally, neural wave equations also dominated the baseline methods in most of the metrics.

In general, this paper introduces a very innovative method which requires further study, and could have huge potential impact to the field.

**Weaknesses:**

There are several weaknesses of this manuscript.

1. The presentation of this manuscript is not optimal. For example, the Figure 1 requires some imagination to understand the design to distinguish the three methods.

2. The experiments are nice, but the neural wave equation might not perform as well as the baseline methods for some data. This could potentially be because the form of the wave equation acts as a strong prior that might not fit the data. It would be nice to understand further the limitation of this framework. The ablation study could potentially be extended.

**Questions:**

1. Can the neural wave equation be applied only to irregularly sampled sequence data, or could it also work in more general sequence modeling settings? What is the reason for only putting this framework into the irregularly sampled sequence data setting?

2. The reviewer suggests improving the quality of Figure 1. Some fonts are too small to read, so please use PDF or SVG formats to enhance the visual clarity. Additionally, could the authors consider redesigning the figure? The current design makes it difficult to distinguish between straight and curved lines. While it makes sense that curved lines represent continuity, there may be a clearer way to convey this distinction.

3. Practically, is this algorithm fast to run?

4. What will be the limitation for incorporating this structure of wave equation. What happens if the real latent dynamics follows another PDE that cannot be modeled as a wave equation?

5. What is the reason that in some experiments, the performance is suboptimal compared to baseline methods?

6. In page 6, Line 302, please consider changing the '-' to ":".

7. The algorithm description of neural wave equation would require a significant rewrite. The current version is hard to be understood.

8. The appendix provides an introduction of the solution of wave equation. The reviewer believes that the authors should provide proper citations here and it is very important.

---

> ### Author Response · Authors · 2024-11-22
> **Response to reviewer HEKw**
>
> First of all, we would like to thank you for appreciating our work. We would also like to sincerely thank you for your detailed review and constructive feedback, which has helped us enhance the presentation of our submission.
>
> **Can the neural wave equation be applied only to irregularly sampled sequence data, or could it also work in more general sequence modeling settings? What is the reason for only putting this framework into the irregularly sampled sequence data setting?**
>
> The Neural Wave Equation can be applied to both regularly and irregularly sampled sequence data. However, we focus on the irregularly sampled data setting because it is one of the most critical applications that can benefit from continuous-time methods. Discrete-time approaches, such as traditional RNNs and LSTMs, often struggle with irregular sampling due to their reliance on fixed time intervals for processing sequences. This limitation can lead to underperformance in scenarios where observations occur at non-uniform intervals, as they fail to effectively capture the temporal relationships between events. In addition to the irregularly sampled sequence data, we have conducted experiments in the regular setting as well.  The preliminary experiment (Fig 2) is done on Etth1 which  contains hourly data of six features (e.g., load, oil temperature) from an electricity transformer. It is a regularly sampled dataset and the neural wave equation outperforms both ODE-RNN and LSTM models.
>
> **The reviewer suggests improving the quality of Figure 1. Some fonts are too small to read, so please use PDF or SVG formats to enhance the visual clarity. Additionally, could the authors consider redesigning the figure? The current design makes it difficult to distinguish between straight and curved lines. While it makes sense that curved lines represent continuity, there may be a clearer way to convey this distinction.**
>
> Taking your advice, we have redesigned the figure 1 to better represent the different approaches. We have used the PDF format for the images in our updated submission to ensure that their resolution is preserved. This choice allows the figures to remain clear and detailed, especially when zoomed in.
>
> **Practically, is this algorithm fast to run?**
>
> Our algorithm is both practically fast and GPU-efficient. We had a discussion of the computational time of Neural Wave Equations in section 5.2 in the original submission. We also provide a speed comparison between most of the existing baselines and our method below and have added the information in **Appendix A.7**
> | Model       | Speed (epoch/s)  |
> |-------------|------------------|
> | CTRNN       | 259              |
> | ODE-LSTM    | 82.39            |
> | CTGRU       | 11.45            |
> | GRUODE      | 11.93            |
> | Avreage     | 91.1925          |
> | BIRNN       | 30.66            |
> | GRUD        | 13.22            |
> | PHASED      | 9.4              |
> | Average     | 17.76            |
> | Neural Wave | 8.66             |
>
> **What will be the limitation for incorporating this structure of wave equation. What happens if the real latent dynamics follows another PDE that cannot be modeled as a wave equation?**
>
> Thank you for raising this insightful question. Theoretically, we agree that in dynamical systems if the latent dynamics follow a different partial differential equation (PDE), the wave equation might not be the most optimal choice. However, the proposed neural wave equation where  the source function is parameterized by a neural network,  allows the model  to adapt to a wide range of underlying dynamics. As a result, the practical performance difference is likely to be minimal, even if the true latent process differs from the wave equation.
>
> Moreover, in real-world sequence labeling problems arising in data science and artificial intelligence, we typically do not have any knowledge of the underlying data generation process. Here, a model can  achieve a good performance if it can learn the representations of the data considering the underlying dependencies among the elements in the sequence.  This is where our choice of neural wave equation based model can be very effective. We chose the wave equation because its FDM discretization reveals that the evolution of hidden states depends on their neighbors, and it includes an additional term which helps to model dependencies which is not present in other PDEs such as the heat equation. More details can be found in the subsection 4.3 Discussions.
>
> Finally, one of the primary objectives of this work was to explore the theoretical rationale for using PDEs in sequence modeling. To this end, we focused on PDEs with known analytical solutions for non-homogeneous cases, specifically the heat and wave equations, as they allowed us to better understand the behavior and capabilities of PDE-based models in a sequence modeling context.

---

> > ### Author Response · Authors · 2024-11-22
> > **Response to reviewer HEKw (continued)..**
> >
> > **What is the reason that in some experiments, the performance is suboptimal compared to baseline methods?**
> >
> > The neural wave equation could act as a strong prior and may not perform as well if the latent dynamics are not continuous. To address this, we included the stance classification experiments, which demonstrate that even when the data is high-dimensional and not generated from continuous dynamics, the neural wave equation does not underperform compared to other SOTA methods.
> >
> > We have also updated the other suggested changes (replacing "-" by ":", citation for the sketches of the analytical solution of PDEs and rewriting the algorithm) in the revised upload of our paper.

---

### Official Review · Reviewer_uqft · 2024-11-04

**Soundness:** 3
**Presentation:** 2
**Contribution:** 3
**Rating:** 6
**Confidence:** 3

**Summary:**

This paper introduces the neural wave equation, a novel approach to handling irregularly sampled sequence data by modeling hidden state evolution using wave equations. The authors propose using a non-homogeneous wave equation with a neural network-parameterized source function to capture sequence dependencies. The method allows continuous modeling across both time and depth dimensions, addressing limitations of existing approaches that use discrete depth transformations. The authors demonstrate the effectiveness of their approach through experiments on various sequence labeling tasks, including person activity recognition, Walker2d kinematic simulation, sepsis prediction, and stance classification, showing competitive or superior performance compared to existing baselines.

**Strengths:**

- Strong theoretical foundation with clear mathematical derivations
- Comprehensive experimental evaluation across diverse datasets
- Thoughtful comparison with existing approaches, especially heat equation-based methods
- Detailed ablation studies showing the importance of different components

**Weaknesses:**

- The motivation for choosing wave equations could be stronger
- Memory consumption issues are noted but not thoroughly addressed
- The connection between theoretical advantages and empirical improvements could be clearer
- Limited discussion of computational complexity trade-offs
- Some implementation details about boundary conditions could be more explicit

**Questions:**

- How sensitive is the model to the choice of wave speed parameter c?
- Could you elaborate on how boundary conditions are handled in practice?
- How does the computational complexity compare to existing methods?
- Have you explored any techniques to reduce memory consumption?
- How does the model perform with very long sequences?

---

> ### Author Response · Authors · 2024-11-22
> **Response to reviewer uqft**
>
> Thank you for taking the time to review our submission and providing valuable feedback. We greatly appreciate your insightful comments and suggestions, which have helped us improve the quality and clarity of our work.
>
> **How sensitive is the model to the choice of wave speed parameter c?**
>
> The wave speed parameter c is a trainable variable in our implementation of the model, and is learnt  via backpropagation.
>
> **Could you elaborate on how boundary conditions are handled in practice?**
>
> In practice, the neural wave equation is solved as an initial value problem with the finite difference discretization scheme. The $h(t,0)$ values are obtained  as the neural network transformation of the input sequence at time t. We use the solvers from torchdyn which assumes
> $\frac{{\partial h(t,0)}}{{\partial d}} = 0$ and $\frac{{\partial h(0,d)}}{{\partial t}} = 0$
> and consequently boundary values are set as $h(t,-1) = h(t,0)$ and $h(-1,d) = h(0,d)$.  **We have also added a detailed description of the boundary conditions and how they are handled in Appendix A.12. It includes a diagram of the grid which will facilitate the understanding of the boundary conditions better.**
> We provide a rough sketch of the boundary conditions that will arise while solving the FDM discretization.
> \begin{equation}
>     h_{t,d+\Delta_{d}} = 2h_{t,d} - h_{t,d-\Delta_{d}} + \frac{\Delta^2_{d}}{\Delta^2_{t}}c^{2}[h_{t+\Delta{t},d} - 2h_{t,d} + h_{t - \Delta_{t},d}]
> \end{equation}
>
> The initial values are needed when $t = 0$ or $d = 0$.
> When $d = 0$,
> $$h_{t,0+\Delta_{d}} =2h_{t,0} - h_{t,0-\Delta_{d}} + \frac{\Delta^2_{d}}{\Delta^2_{t}}c^{2}[h_{t+\Delta_{t},0}-2h_{t,0}+h_{t-\Delta_{t},0}]$$
> The $h_{t,0-\Delta_{d}}$ value is not available to us and we use $h_{t,0-\Delta_{d}} = h_{t,0}$ in our implementation.
> When $t = 0$,
> $$h_{0,d+ \Delta_{d}} = 2h_{0,d} - h_{0,d- \Delta_{d}} + \frac{\Delta^2_{d}}{\Delta^2_{t}}c^{2}[h_{0 + \Delta_{t},d}-2h_{t,d}+h_{0 - \Delta_{t},d}]$$
> Here, $h_{0-\Delta_{t},d}$ is again not known to us and we use $h_{0-\Delta_{t},d} = h_{0,d}$ in our implementation.
>
> **How does the computational complexity compare to existing methods?**
>
> We had provided  a comparison of the time complexity of the proposed approach with some baselines in the experiments section (Section 5.2, Lines 457-458) of the paper. Here, we compared the time complexity of our model with neural CDE and CDR-NDE models. We had also discussed about the memory complexity under the Ablation study (Section 5.5, lines 523-526). In addition to these, we have now conducted a more thorough analysis of the computational complexity (both time and space) including other  baselines as well. We provide the results on the Walker dataset below.  **We have also added a separate section on computational complexity in the Appendix (A7).**
> | Model       | Memory (in MB) | Speed (epoch/s)  |
> |-------------|----------------|------------------|
> | CTRNN       | 321            | 259              |
> | ODE-LSTM    | 348            | 82.39            |
> | CTGRU       | 662            | 11.45            |
> | GRUODE      | 161            | 11.93            |
> | Avreage     | 373            | 91.1925          |
> | BIRNN       | 162            | 30.66            |
> | GRUD        | 48             | 13.22            |
> | PHASED      | 38             | 9.4              |
> | Average     | 82.67          | 17.76            |
> | Neural Wave | 1972           | 8.66             |
>
> It is important to emphasize that the observed speed-memory tradeoff arises from our implementation technique rather than being an inherent property of the model. Specifically, in most RNN variants with ODE solvers, it is necessary to loop over the sequence dimension because ODE solvers typically cannot handle 3-dimensional data directly. (Please refer to Appendix A.6 for more details, as this is not a common implementation for such models.) To address this limitation, we collapsed the batch and sequence dimensions into a single dimension. This approach enabled us to utilize the GPU more efficiently, significantly improving speed by eliminating the need for looping. However, this optimization leads to a higher peak memory allocation, as the entire collapsed batch-sequence matrix must fit in memory during computation.
>
> **Have you explored any techniques to reduce memory consumption?**
>
> We were more concerned about the speed than memory consumption since it is a major problem we faced while running ode-rnn methods. We would like to explore better methods to reduce memory consumption in future work, for instance computation of adjoint backpropagation for 2nd order solvers[1].
>
> [1] Liu et al, Second-order neural ODE optimizer,NIPS'21: Proceedings of the 35th International Conference on Neural Information Processing Systems

---

> > ### Author Response · Authors · 2024-11-22
> > **Response to reviewer uqft (continued)..**
> >
> > **How does the model perform with very long sequences?**
> >
> > To check how the model performs with very long sequences, we came up with a creative experiment. We take the Ettm1 data - which contains data of six features (e.g., load, oil temperature) from an electricity transformer every 15 minutes. We keep the prediction window same (24 time steps) and increase the input sequence gradually (96,138,336) to see how the model performs with increasingly long sequences of data.
> >
> > |                | Hidden Dimension |         |
> > |----------------|------------------|---------|
> > | Input Sequence | 32               | 64      |
> > | 96             | 0.2002           | 0.1564  |
> > | 138            | 0.2211           | 0.1497  |
> > | 336            | 0.2314           | 0.1714  |
> >
> > As the input sequence length increases, the model's performance tends to degrade, as evident from the results presented in the table. Specifically, we observe a slight but consistent increase in error when moving from shorter sequences (96) to longer ones (336). This behavior aligns with theoretical insights into the relationship between the input sequence length and the dimensionality of the hidden state.[1]
> >
> > The performance drop with increasing sequence length can be  mitigated by increasing the hidden state size. For example, initially with hidden size 32, the sequence length of 96 (MSE = 0.2002) outperforms the sequence length of 138 (MSE = 0.2211). But on increasing the hidden size to 64, the sequence length of 138 (MSE = 0.1497) outperforms sequence length of 96 (MSE = 0.1564). In all 3 different sequence lengths, error decreases significantly with increasing the hidden size.
> > This observation supports the relationship described in the Zoology paper [1], where the hidden state dimension needs to scale approximately linearly with the input sequence length to maintain comparable performance.
> >
> > To further clarify, the model's performance depends on its capacity to adequately capture the underlying dependencies within longer sequences. This capacity is governed by the hidden state dimension. By increasing the hidden state size, the model can retain and process more information over longer sequences, which mitigates the degradation in performance.
> >
> > [1]Simran Arora · Sabri Eyuboglu · Aman Timalsina · Isys Johnson · Michael Poli · James Y Zou · Atri Rudra · Christopher Re, Zoology: Measuring and Improving Recall in Efficient Language Models, ICLR 2024
> >
> >
> > We hope that our revisions have resolved the issues you raised. If you find our responses satisfactory and the changes adequate, we kindly ask you to consider reflecting this in your overall evaluation.

---

> > > ### Author Response · Authors · 2024-11-26
> > >
> > > Thank you once again for taking the time to review our submission. We understand that the review process is demanding, and we truly appreciate the effort and time you dedicate. However, we noticed that there hasn’t been a response to our rebuttal or clarification. We would kindly like to follow up to see if there is anything further we can clarify or provide to assist in the review process. We are happy to address any concerns you may have regarding the rebuttal.
> > >
> > > If you find our responses satisfactory and the changes adequate, we kindly ask you to consider reflecting this in your overall evaluation.

---

> > > > ### Comment · Reviewer_uqft · 2024-11-26
> > > >
> > > > Thank you for your detailed and thoughtful response to my review. I appreciate the effort you have put into addressing my concerns and providing additional explanations and experimental results.
> > > >
> > > > Your clarification that c is a trainable parameter learned via backpropagation is appreciated. This is a reasonable design choice that likely enhances adaptability to diverse datasets. However, a sensitivity analysis demonstrating how \(c\)'s learned values impact performance across tasks would further strengthen this point. Including such an analysis in future work or revisions would be helpful.
> > > >
> > > > Your explanation of how boundary conditions are handled, supported by a diagram in Appendix A.12, effectively clarifies this aspect of the model. This is a solid response to the concern, and the additional details make the implementation clearer. However, it would be beneficial to discuss whether the choice of boundary conditions limits generalizability to other sequence types or systems.
> > > >
> > > > The detailed comparison of memory and speed metrics and your explanation of the trade-off between memory usage and computational efficiency are appreciated. While the transparency of your approach is commendable, the lack of immediate strategies to reduce memory usage is a notable limitation.
> > > >
> > > > The experiments on varying sequence lengths provide useful insights, particularly the relationship between hidden state size and performance degradation. This analysis demonstrates that the model can handle long sequences with appropriate scaling.
> > > >
> > > > Your effort to improve the paper’s clarity, including updates to figures, additional experimental results, and revised sections, is apparent and appreciated.
> > > >
> > > > Your responses have resolved many of my concerns and clarified the contributions of the work. While some areas still require further exploration (e.g., memory efficiency, sensitivity analysis), the quality of the rebuttal and additional experiments warrant an updated recommendation.
> > > >
> > > > Updated Score: 6

---

> > > > > ### Author Response · Authors · 2024-11-29
> > > > > **Update on memory usage**
> > > > >
> > > > > Thank you for your thoughtful feedback, particularly regarding the memory overhead.
> > > > >
> > > > > We would like to share some updates related to the points you raised. Taking your insights into account, we conducted further research into strategies to handle memory usage more effectively.
> > > > >
> > > > > We used an engineering trick called **gradient checkpointing** on the Walker 2d Kinematics dataset. Through gradient checkpointing [1,2] the MLP that is passed through the solver, we significantly reduce memory usage (from 1.9 GB to 400 MB) while maintaining reasonable training times (a modest increase from 8 seconds to 12 seconds per epoch). While this is still a tradeoff, this approach provides better control over the balance between memory efficiency and computational cost.
> > > > > We sincerely appreciate your feedback, which guided us in strengthening this aspect of our work.
> > > > >
> > > > > [1]https://github.com/cybertronai/gradient-checkpointing
> > > > >
> > > > > [2] https://pytorch.org/docs/stable/checkpoint.html
> > > > >
> > > > > Sensitivity of c
> > > > >
> > > > > As part of our efforts to address your suggestion regarding sensitivity analysis for c, we conducted a small study on the homogeneous model (without the neural network component since the neural network likely compensates for suboptimal initial values of c) on the walker dataset. This allowed us to isolate the impact of c without the corrective capabilities of a trainable neural network. The results (as shown in the table) indicate that test loss varies significantly with c.
> > > > >
> > > > > | C   | Test Loss  |
> > > > > |-----|------------|
> > > > > | 0.1 | 2.728      |
> > > > > | 0.5 | 2.429      |
> > > > > | 1   | 1.281      |
> > > > > | 3   | 2.329      |
> > > > > | 5   | 6.243      |
> > > > >
> > > > > For the model where c is trainable, the MSE is 0.99 (reported in the ablation study). This suggests that the flexibility of a trainable c mitigates sensitivity, improving overall robustness.
> > > > >
> > > > > Boundary Conditions
> > > > >
> > > > > We appreciate your observation regarding the choice of boundary conditions and its potential impact on generalizability. While our current approach effectively handles the tested sequence types, the boundary conditions are influenced by the limitations of the underlying solver. In future work, we aim to explore improvements to solvers to better accommodate diverse systems and sequence types, potentially enhancing generalizability further.

---

### Author Response · Authors · 2024-11-22

**We are deeply grateful to the reviewers for their effort and time spent in reviewing our work. We value their critical and insightful comments that have greatly helped us improve the quality of our work. Furthermore, we appreciate the suggestions that have enabled us to improve the organization and the presentation of our paper.**

**Changes Made**

1. We have revised the introduction and section 3.3 for better readibility and highlighting why PDEs or in particular wave equation is suitable for sequence modeling.

2. We have resdesigned the figure 1.

3. We have changed t' with d (for representing the depth direction) for better readibility of the equations.

4. We have grouped the baselines in table 1 as per categories of figure 1.

5. We have added Appendix A.12 with detailed information on how boundary value conditions are handled in our method.

6. We have added Appendix A.7 with further comparison on computational complexity of neural wave equation with the baselines.

---

### Meta-Review · Area_Chair_M6ho · 2024-12-22

**Metareview:**

The paper proposes the neural wave equations, a novel approach for handling sequence data, especially irregularly sampled sequences, by modelling hidden states evolution using a wave equation parameterized by a neural network. Reviewers appreciated the innovative idea with a strong theoretical foundation, comprehensive experiments, thoughtful comparison with existing approaches, and detailed ablation studies. The reviewers raised concerns about presentation clarity (including motivation for using the wave equations and figures and algorithms lack clarity), the need for better contextualization with related work and the method's limitations, high memory consumption compared to baselines, limited discussion of computational complexity trade-offs, and the methods reliance on the wave equation as a prior, which may not align with the dynamics of all datasets. The authors addressed these concerns during the rebuttal and discussions by refining sections and figures, adding new experiments (particularly on sensitivity analysis of the wave speed parameter), partially addressing memory concerns by applying gradient checkpointing, and providing detailed explanations in the appendix. Despite some unresolved issues, such as fully addressing memory efficiency and further sensitivity and ablations studies, the reviewers unanimously recommend accepting the paper based on its novelty and strong empirical results, and therefore, I recommend accepting the paper.

**Additional Comments On Reviewer Discussion:**

A summary of the rebuttal and discussion is described in the meta-review, as well as arguments for the final decision.

---

### Decision · Program_Chairs · 2025-01-22

Accept (Poster)